



# Nine years of SMOS Sea Surface Salinity global maps at the Barcelona Expert Center

Estrella Olmedo[1], Cristina González-Haro[1], Nina Hoareau[1], Marta Umbert[1],
Verónica González-Gambau[1], Justino Martínez[1], Carolina Gabarró[1], and Antonio Turiel[1]

[1]Barcelona Expert Center (BEC) and Institute of Marine Sciences (ICM), CSIC, P. Marítim de la Barceloneta, 37-49, 08003 Barcelona, Spain

**Correspondence:** Estrella Olmedo (olmedo@icm.csic.es)

**Abstract.**

After more than 10 years in orbit, the Soil Moisture and Ocean Salinity (SMOS) European mission is still a unique, high-quality instrument for providing Soil Moisture over land and Sea Surface Salinity (SSS) over the oceans. At the Barcelona Expert Center (BEC), a new reprocessing of 9 years (2011-2019) of global SMOS SSS maps has been generated. This work presents the algorithms used in the generation of BEC global SMOS SSS product v2.0, as well as an extensive quality assessment. Three SMOS SSS fields are distributed: a high-resolution level 3 product (with doi: http://dx.doi.org/10.20350/digitalCSIC/12601 (Olmedo et al., 2020a)) consisting of a binned SSS in 9-day maps at $0.25 \times 0.25^o$; a low-resolution level 3 SSS computed from the binned salinity by applying a smoothening spatial window of 50-km radius; and a level 4 SSS (with doi: http://dx.doi.org/10.20350/digitalCSIC/12600 (Olmedo et al., 2020b)) consisting of daily, $0.05 \times 0.05^o$ maps that are computed by multifractal fusion with Sea Surface Temperature maps. For the validation of BEC SSS products, we have applied a battery of tests aiming at the assessment of quality of the products both in value and in structure. First, we have compared BEC SSS products with near-to-surface salinity measurements provided by Argo floats. Secondly, we have assessed the geophysical consistency of the products characterized by singularity analysis, and also the effective spatial resolutions are estimated by means of Power Density Spectra and Singularity Density Spectra. Finally, we have calculated full maps of SSS errors by using Correlated Triple Collocation. We have compared the performance of BEC SMOS product with other satellite SSS and reanalysis products. The main outcomes of this quality assessment are: i) the bias between BEC SMOS and Argo salinity is lower than 0.02 psu at global scale, while the standard deviation of their difference is lower than 0.34 and 0.27 psu for the high and low resolution level 3 fields (respectively) and 0.24 psu for the level 4 salinity; ii) the effective spatial resolution is around 40 km for all SSS products and regions; and iii) BEC SMOS level 4 product is globally the one with the lowest salinity error, while BEC SMOS low-resolution level 3 more accurate in regions strongly affected by rainfall and continental freshwater discharges.

*Copyright statement.* TEXT





## 1 Introduction

The European Space Agency (ESA) Soil Moisture and Ocean Salinity (SMOS) satellite was launched in November 2009, carrying the first orbiting radiometer that collects regular and global observations from space of two Essential Climate Variables

(ECV) according to the Global Climate Observing System: Sea Surface Salinity (SSS) and Soil Moisture (SM) (Font et al., 2010; Kerr et al., 2010; Mecklenburg et al., 2009). After more than 10 years in orbit, the SMOS mission has been a success in terms of both technology and science, providing SSS and SM data derived from the SMOS measurements (Reul et al., 2020; Ker, 2016) (and references therein).

The Barcelona Expert Centre (BEC) was created in 2007 to support the Spanish contribution to SMOS mission activities.

Since the beginning, BEC's goals have been to contribute to the quality assessment and the development of algorithms for the retrieval of geophysical variables from SMOS data as an ESA Level 2 Ocean Salinity Expert Support Laboratory; and to the calibration and validation activities as a Level 1 Expert Support Laboratory. In the recent years, BEC has developed a SMOS SSS internal processing chain that generates SSS maps from SMOS raw data (Level 0) to Level 3 and 4 (L3 and L4 added-value SSS maps), thus allowing the integration of improvements in the different levels of the processing. The resulting products are

freely distributed through a sftp service (http://bec.icm.csic.es/bec-ftp-service/).

In this work we present the new reprocessing of the BEC SMOS SSS global L3 and L4 products v2.0 for a 9-years period comprising 2011 to 2019, that comes with an improvement of the currently used methodology. This new reprocessing is focused on four aspects:

- Improving salinity gradients: A new filtering criterion that is more geophysically consistent has been introduced.

- Improving the latitudinal and seasonal biases: An empirical correction to reduce the latitudinal and seasonal biases that affected the previous version of the product (Olmedo et al., 2019b) has been applied.

- Improving the quality of the acquisitions close to the coast: The interpolation scheme and also the level 4 fusion techniques have been adapted to preserve small-scale gradients close to the coast.

- Providing an estimate of the sea surface salinity uncertainty: An explicit expression to propagate the errors from bright-

ness temperature uncertainties to the final SSS product have been introduced.

To assess the performance of the BEC SMOS SSS product v2.0, the complete 9-year time series of SSS maps is first compared with the salinity measurements provided by Argo. Secondly, an extensive battery of validation methods is applied to one year (2017) of data, and the results are compared with three other satellite and one reanalysis SSS products. Those methods are: i) statistics of the differences with Argo salinity match ups; ii) singularity analysis to assess the geophysical

consistency of the data (Turiel et al. (2008b)); iii) spectral analysis to analyse the effective spatial resolution of each product, using power density spectra (PDS) and singularity power spectra (SPS) (Hoareau et al. (2018b)), and iv) triple collocation analysis to estimate the errors of the different products (González-Gambau (2020)).

This paper is structured as follows. Section 2 describes how BEC SMOS SSS product v2 is generated: subsection 2.1 introduces the datasets used in the generation of the product and subsection 2.2 describes the algorithm itself. Section 3



presents the quality assessment of the data: subsection 3.1 presents the different datasets used for comparison and validation; subsection 3.2 describes the applied methods and associated metrics; and subsection 3.3 presents the results of the validation exercise. Finally, the conclusions are summarized in Section 4.

## 2   Generation of the BEC SMOS SSS global product v2.0

### 2.1   Data sets used in the generation of the product

### 2.1.1   SMOS Brightness Temperature

The Brightness temperatures (TBs) obtained from SMOS MIRAS L1B v620 product provided by ESA are used as the input for the SMOS SSS retrieval. This data set is freely available at: https://earth.esa.int/eogateway/catalog/smos-science-products?text=SMOS+Britghness+Temperatures. The L1B v620 product contains the Fourier coefficients of the measured brightness temperatures. Starting from this product, using ESA's Earth Observation Customer Furnished Item (EOCFI) orbit propagation libraries (ESA, 2014) and following a similar procedure as the one used in the operational SMOS level 1 processor chain (Deimos, 2014), the measured TBs are obtained in the antenna reference frame (ARF). The unique difference from the standard processor is the number of points contained per snapshot. The operational processor uses, at antenna level, an hexagonal grid of $128 \times 128$ points. The projection of this antenna grid into the ground provides a nominal resolution of about 15 km at bore-sight. This resolution is more than twice the SMOS theoretical finer resolution (McMullan et al., 2008). Thus, we reduce 70   the computational cost without actually losing information by using an antenna hexagonal grid of $64 \times 64$ points for a 30-km resolution at boresight.

### 2.1.2   Sea Surface Temperature

The Operational Sea Surface Temperature and Sea Ice Analysis (OSTIA) (see Donlon et al. (2012)) maps are used as template in the generation of the Level 4 SSS v2.0 products (see 2.2.7). OSTIA Sea Surface Temperature (SST) results from the combination of satellite data provided by the Group for High Resolution Sea Surface Temperature (GHRSST) project, combined with in situ observations. The analysis product is obtained using a variant of the optimal interpolation (OI) method described in (Martin et al., 2007) at a spatial resolution of $0.05°$ (approx. 5km) and daily frequency. OSTIA SST data are provided in netCDF format every day and it is freely available at the Copernicus Marine Environment Monitoring Service (CMEMS) service desk (https://resources.marine.copernicus.eu/?option=com_csw&task=results?option=com_csw&view=details&product_80   id=SST_GLO_SST_L4_NRT_OBSERVATIONS_010_001).

### 2.1.3   Auxiliary data used in the salinity retrieval

The auxiliary data used for the SSS retrieval are provided by the European Centre for Medium range Weather Forecast (ECMWF) (Freitas, 2013; Sabater and De Rosnay, 2010). For each satellite overpass, an ECMWF auxiliary file co-located in time and space with SMOS is provided by ESA. The following fields are used for the retrieval: sea ice cover, sea surface





temperature, rain rate, wave model, 10-meter wind speed, 10-meter neutral equivalent wind (zonal and meridional compo-
nents), significant height of wind waves, 2-meter air temperature, surface pressure, and vertically integrated total water vapour
(Zine et al., 2007).

We have used as multiyear salinity reference the annual climatological salinity value provided by the World Ocean Atlas
2013 (WOA2013) at $0.25° \times 0.25°$ (Zweng et al., 2013). The SSS provided by WOA2013 is taken as the reference value to

be added to SMOS salinity anomalies (see section 2.2.1). We use the average decadal product, which is accessible at Na-
tional Oceanographic Data Center (https://www.nodc.noaa.gov/cgi-bin/OC5/woa13/woa13.pl). We have also used the monthly
climatology at $0.25° \times 0.25°$ provided by WOA2013 for the correction of latitudinal and seasonal biases (see section 2.2.5).

## 2.2 Algorithm description

### 2.2.1 Retrieval of SMOS debiased Sea Surface Salinity Anomalies

The debiased non-Bayesian (DNB) retrieval approach proposed in Olmedo et al. (2017) has been used to retrieve the 9 years
(2011-2019) of SMOS SSS maps. This methodology consists of retrieving a single value of SSS from each First Stokes Bright-
ness Temperature (TB) measurement, that we refer to as raw SSS retrieval. These raw SSS are then appropriately classified,
filtered and combined to build global SSS maps.

For the retrieval of raw SSS, the difference between the First Stokes TB measured and modelled is optimized as a function

of the salinity value. A geophysical forward model links the modeled TB to the SSS. Besides the dielectric constant model
proposed by Klein and Swift (1977) for TB coming out flat sea, the forward model accounts for the contribution to TB of the
sea surface roughness (Guimbard et al., 2012), the reflected emission of the atmosphere, the reflection on sea surface of the
galactic emission (Tenerelli et al., 2008) and the sun glint (Reul et al., 2007). All these contributions are taken into account in
the salinity retrieval.

All the raw salinity retrievals over 9 years ($s_n^{raw}$ with $n = 1, \ldots, N$ where $N$ is the total number of raw retrievals in 9 years)
are classified as a function of the satellite overpass direction ($d$), latitude ($\varphi$), longitude ($\lambda$), across-track distance ($x$), and
incidence angle ($\theta$). The underlying hypothesis of this approach is that the systematic errors (i.e., those which are independent
of time) are the same for all the $s_n^{raw}$ that are acquired under each fixed condition $\gamma = (\varphi, \lambda, d, x, \theta)$. Therefore, the systematic
errors are the same for all the retrievals in the set $\{s_n^{raw}(\gamma)\}$ with $n = 1, \ldots N_\gamma$ and $N_\gamma$ the number of retrievals during 9 years

with that specific value of $\gamma$.

We have defined an estimator of the "typical value" or central estimator of the ensemble $\{s_n^{raw}(\gamma)\}$, that we will call SMOS-
based climatology, $s^c(\gamma)$. We want, by construction, this SMOS-based climatology to represent the sum of a multiyear mean
salinity value (which is a geophysical property) and the bias associated to that particular tuple $\gamma$ (which is of instrumental
origin and we want to remove). Therefore, the SMOS-based climatologies can be used for correcting all the retrievals in the set

$\{s_n^{raw}(\gamma)\}$. We have used the same central estimator than the one proposed in Olmedo et al. (2017), that is, the mean around
the mode in an interval of $\pm\sigma_\gamma$ (the standard deviation of $\{s_n^{raw}(\gamma)\}$).





Then, we have computed SMOS debiased SSS anomalies ($\{s'_n(\gamma)\}$) by subtracting to each individual $s_n^{raw}(\gamma)$ the corresponding SMOS-based climatology $s^c(\gamma)$.

Finally, the debiased salinity value for the acquisition conditions $\gamma$, $s_n(\gamma)$, is computed by adding an external multiyear
salinity reference to the $s'_n(\gamma)$; in this case, we have used the annual salinity field provided by WOA2013.

The retrieval algorithm proposed above effectively removes local biases, especially those produced by the land-sea contamination and artifacts produced by permanent Radio Frequency Interference (RFI) sources.

### 2.2.2 Estimation of SSS error

Each value of raw salinity $s_n^{raw}$ can be associated a retrieval error which is computed according to the next equation:

$$\epsilon_n = \frac{1}{2} \frac{\sqrt{(\sigma_n^H)^2 + (\sigma_n^V)^2}}{\frac{\partial I_n}{\partial s}} \tag{1}$$

where the $\sigma_n^H$ and $\sigma_n^V$ are the radiometric sensitivities for the horizontal (H) and vertical (V) polarizations of the brightness temperature, respectively (that are contained in the ESA L1B product), and $\frac{\partial I_n}{\partial s}$ is the derivative of the modelled First Stokes divided by 2 ($I_n$) with respect to the salinity (that can be estimated numerically).

### 2.2.3 Filtering criteria

Filtering out degraded measurements in the generation of the SMOS SSS maps is a key aspect. Without applying any filter, the error may become too large for many scientific applications; on the other hand, when the filtering criteria are too strict, the coverage of maps may be dramatically decreased and part of the geophysical variability may be lost. In Olmedo et al. (2017), filtering criteria based on the statistical properties of $\{s_n^{raw}(\gamma)\}$ were proposed and the resulting maps led to an almost complete coverage with an acceptable salinity accuracy (see Olmedo et al. (2017) for more details). We revisit these filtering
criteria in order to decrease the error of the retrieved salinity and improve the description of salinity gradients in highly dynamic regions.

We apply the following filtering criteria:

- **Basic filtering**: Any $s_n^{raw}(\gamma)$ out of the range of [0, 50] psu is not considered as part of the corresponding set of valid $\{s_n^{raw}(\gamma)\}$.

- **Discarding some full sets of** $\{s_n^{raw}(\gamma)\}$: For a given value of $\gamma$, we consider a particular set of $\{s_n^{raw}(\gamma)\}$ valid only when:

  - It contains more than 100 salinity retrievals;

  - The standard deviation of its distribution is lower than 10 psu;

  - The absolute value of the skewness of the distribution is lower than 1; and

- The kurtosis of the distribution is greater than 2.





- **Outlier criteria**: We discard specific salinity retrievals $s_n^{raw}(\gamma)$ when the corresponding SMOS debiased salinity anomaly $(s_n'(\gamma))$ satisfies:

$$|s_n'(\gamma)| = |s_n^{raw}(\gamma) - s^c(\gamma)| > \sqrt{\sigma_\gamma^2 + 25\sigma_{\varphi,\lambda}^2} \qquad (2)$$

where $\sigma_{\varphi,\lambda}^2$ is the geophysical variance of the salinity expected at the gridpoint $(\varphi,\lambda)$. In order to estimate $\sigma_{\varphi,\lambda}^2$ we use SMOS 9-day salinity maps that are computed from a more relaxed filtering criteria, which is:

$$|s_n^{raw}(\gamma) - s_\gamma^c| < 2\sigma_\gamma. \qquad (3)$$

Notice that $\sigma_\gamma$ is always greater than $\sigma_{\varphi,\lambda}$, because $\sigma_\gamma$ contains the variability corresponding to the salinity uncertainty and the salinity geophysical variability (see Olmedo et al. (2019a)).

- **Temporal and geophysical consistency**: We temporally and spatially collocate all the debiased retrievals $s_n(\gamma)$ in 9-day maps with the fixed grid at $0.25^o \times 0.25^o$. The resulting collocated set of SSS is denoted as $\{s(t_T,\varphi,\lambda)\}$, being $t_T$ all the acquisition times in the 9-day period which is indexed by the time $T$ (typically, $T$ corresponds to the central day). In particular, for a given geographical location $(\varphi,\lambda)$, we combine all the different values of SSS under all acquisition conditions at that specific location and 9-day period, what means combining all the satellite overpass directions $(d)$, across-track distances $(x)$, and incidence angles $(\theta)$ that happen at all the time $t_T$ in that period. Then, we consider as valid salinity measurements only those satisfying:

$$|s(t_T,\varphi,\lambda) - \bar{s}(T,\varphi,\lambda)| < \sigma(T,\varphi,\lambda), \qquad (4)$$

with $\bar{s}(T,\varphi,\lambda)$ and $\sigma(T,\varphi,\lambda)$ being the mean and standard deviation of the set $\{s(t_T,\varphi,\lambda)\}$, that is, all values of SSS at longitude $\varphi$, latitude $\lambda$ and 9-day period centered around $T$. Finally, we average the valid salinity values of $\{s(t_T,\varphi,\lambda)\}$ in that period to obtain the binned 9-day map at $0.25^o \times 0.25^o$, $s_0(T,\varphi,\lambda)$.

### 2.2.4 Mitigation of temporal biases

SMOS measurements are affected by biases that depend on time (see Martín-Neira et al. (2016)). The methodology described in 2.2.1 aims at removing the systematic biases affecting SMOS measurements, i.e., those biases that depend on the acquisition conditions $(\gamma)$ but not on time. To address the temporal biases, we follow the approach proposed in Olmedo et al. (2017), which consists of assuming that the global average of SSS does not change with time. Figure A1 shows the difference between the spatial averaged salinity value of $s_0(T,\varphi,\lambda)$ and the spatial averaged salinity value of the annual reference (WOA13). We assume that for a given time $T$ this difference has to be zero. Therefore, we correct each map with this difference, imposing the spatial averaged salinity value of every global map to be equal to the annual reference. We notate the temporal corrected binned salinity fields as $s_1(T,\varphi,\lambda)$.



### 2.2.5 Correction of latitudinal-seasonal biases

The corrections applied so far aim at systematic biases which are time-independent or space-independent, and therefore can be corrected separately. However, after applying both corrections, residual biases depending at the same time on time and on the geographical position are still present (see Figure A2). These latitudinal-seasonal biases are known to happen also in other L-band satellite missions and are supposed to be due to the different direct influence of the Sun on the instrument along its trajectory depending on the season of the year. We have therefore applied the latitudinal-seasonal bias correction proposed in

Olmedo et al. (2019b), that is computed as follows:

- We compute SMOS monthly climatologies $\bar{s}_1(m, \varphi, \lambda)$ for each month $m$ of the year by averaging all the $s_1(T, \varphi, \lambda)$ where $T$ belongs to the same month $m$ of the processed nine years. Recall that this climatology is defined on a $0.25^o \times 0.25^o$ grid, as this is the grid for $s_1(T, \varphi, \lambda)$.

- For each month $m$, we subtract the WOA2013 monthly climatology, $s^{WOA}(m, \varphi, \lambda)$, from the corresponding SMOS

monthly climatology:

$$\Delta s_1(m, \varphi, \lambda) = \bar{s}_1(m, \varphi, \lambda) - s^{WOA}(m, \varphi, \lambda)$$

- We fit $\Delta s_1(m, \varphi, \lambda)$ by a second degree polynomial of the latitude. That is, for every month, $m$, and every value of latitude in the $0.25^o$ grid, $\varphi$, we compute the polynomial $p(m, \varphi)$

$$p(m, \varphi) = a(m)\varphi^2 + b(m)\varphi + c(m)$$

that minimizes the following cost function:

$$\sum_{\lambda} \left( \Delta s_1(m, \varphi, \lambda) - p(m, \varphi) \right)^2,$$

- After computing the optimal polynomials $p(m, \varphi)$, we correct the maps $s_1(t, \varphi, \lambda)$ by daily interpolating the polynomial $p(m, \varphi)$ to the specific moment of the month. We denote the latitudinal-seasonal debiased SSS by $s_2(T, \varphi, \lambda)$.

### 2.2.6 Mitigation of residual spatial biases

After applying all the above corrections, we make the last check. By construction, at each geographical location the average salinity of the full period should be equal to the multiyear reference introduced in section 2.2.1. We have found significant differences between both averages that may be due to an inaccurate determination of the SMOS-based climatology. This may happen when the distributions of the values $\{s_n^{raw}(\gamma)\}$ for a given $\gamma$ significantly deviates from a Gaussian distribution (especially, if it is slightly skewed), and then differences between the modes (used in the computation of $s^c(\gamma)$) and the means

of all $s_n(\gamma)$ after applying the filtering criteria (section 2.2.3) are significant. In order to mitigate this last bias, we remove the map corresponding to the difference between the mean average of all the SMOS SSS in the 2011-2019 period ($s_2(T, \varphi, \lambda)$) and WOA2013 (see Figure A3). The resulting salinity field is our L3 high-resolution product $s^{L3}(T, \varphi, \lambda)$. In future versions of this product we will introduce a better definition of the SMOS-based climatology to avoid this last correction step.





### 2.2.7 Multifractal fusion techniques

We use the multifractal fusion techniques introduced in Umbert et al. (2014) to increase the spatial and temporal resolutions of SMOS SSS L3 maps (Olmedo et al., 2016). Multifractal fusion methods are based on the hypothesis that different ocean scalars have the same singularity exponents (SE). From a mathematical point of view, the SE of a function at a given point is a measure of the local regularity of the function at that point (Turiel et al., 2008a). It has been shown that synoptic maps of different ocean scalars show the same multifractal structure due to the effect of geophysical turbulence. Starting from the SE

extracted from SST maps (Turiel et al., 2005, 2008b), it has been observed that other scalars, such as chlorophyll concentration maps (Nieves et al., 2007; Umbert et al., 2020) and even brightness temperature maps at given frequencies (Isern-Fontanet et al., 2007) present the same structure and even values of SE. This correspondence can be used to improve the quality of the SMOS SSS L3 maps by using as a template OSTIA SST, which is an ocean scalar measured with better spatio-temporal resolution and quality than SSS. Assuming that both variables have the same SE, it can be shown (Umbert et al., 2014) that as

a first order approximation the following local relationship holds:

$$SSS = a \cdot SST + b \tag{5}$$

where $a$ and $b$ are smooth functions, that is they must have small gradients, as otherwise they would introduce additional SE. The estimation of the smooth functions $a$ and $b$ is done by means of a local weighting average (see Umbert et al. (2014) and Olmedo et al. (2016) for more details). Taking advantage of the fact that $a$ and $b$ do not have sharp variations over large

regions, the evaluation of $a$ and $b$ is performed by locally-weighted linear regression. We have employed a similar local-weighting function as in Olmedo et al. (2016), that is, the inverse of the 4-th power of the distance to the central point. In order to better describe the small-scale features, the local weighting is limited to points at most at a distance of $R = 2.5°$ from the central point.

By means of this multifractal fusion method, SMOS L4 SSS maps with the same spatial and temporal resolutions as the

template (OSTIA SST), i.e., daily maps at a spatial grid of $0.05° \times 0.05°$, are obtained.

### 2.2.8 BEC SMOS SSS global product v2.0

The BEC SMOS SSS L3 global product v2.0 consists of 9-day SSS maps at a regular grid of $0.25^o \times 0.25^o$ generated daily. The product is distributed in netCDF files and it contains two different salinity fields and one estimation of the SSS uncertainty. The two SSS fields are denoted as:

– BEC SMOS HR SSS product: where HR stands for High Resolution and contains the binned salinity field $s^{L3}(T, \varphi, \lambda)$.

– BEC SMOS LR SSS product: where LR stands for Low Resolution and is a low pass filtered version of $s^{L3}(T, \varphi, \lambda)$, computed by applying a spatial radio of 50km. This product is denoted by $s_{low}^{L3}(T, \varphi, \lambda)$.

The BEC SMOS L4 SSS product v2.0 (hereafter BEC L4) consists of daily SSS maps at a regular grid of $0.05^o \times 0.05^o$. This product is denoted by $s^{L4}(T, \varphi, \lambda)$.





## 3  Quality assessment

### 3.1  Data sets for validation

#### 3.1.1  Satellite Sea Surface Salinity

We have compared the performance of the new BEC products with that of other satellite SSS products. We have centered the validation in the year 2017 because there is not any large-scale geophysical phenomenon (such as El Niño or La Niña events), and also because SSS products produced by National Aeronautics and Space Administration (NASA)'s Soil Moisture Active Passive (SMAP) mission are available (this mission has been operating since early 2015 (Entekhabi et al., 2010)). The satellite SSS products used for the intercomparison are:

- CATDS SMOS products: 9-day SMOS SSS maps provided by Centre Aval de Traitement des Données SMOS (CATDS). We use the L3 debiased v4 freely available at: http://catds.ifremer.fr/Products/Available-products-from-CEC-OS/Locean-v2019. This product decreases the mean bias over the open ocean with respect to previous versions and it improves ice filtering, which leads to an improvement of SSS at high latitudes, especially in the Southern Ocean (Boutin et al., 2016, 2018; Kolodziejczyk et al., 2016).

- JPL SMAP products: 8-day SMAP SSS maps provided by Jet Propulsion Laboratory (JPL). We use the Level 3 version 4.2 freely available at: https://podaac-opendap.jpl.nasa.gov/opendap/allData/smap/L3/JPL/V4.2/. Updates of version 4.2 with respect to previous versions include: improvement in the TB calibration using an adjusted reflector emissivity, the inclusion of a SST dependence on a flat surface emissivity model, use of updated land correction tables, and inclusion of averaged ice concentration ancillary data (Fore et al., 2016).

- REMSS SMAP products: 8-day running Remote Sensing Systems SMAP Level 3 Sea Surface Salinity Standard Mapped Image version v4 which is freely available at www.remss.com/missions/smap. In particular, we have used the field sss_smap, which is a smoothened measurement at approximately 70km resolution. The major change in Version 4.0 from Version 3.0 is an improved land correction, which allows for SMAP salinity retrievals closer to the coast (Meissner et al., 2018).

#### 3.1.2  In situ salinity: Argo floats

For the purpose of direct comparison of values, we have used in situ salinity data obtained by Argo profilers. We consider the uppermost Argo salinity between 5 and 10 m depth (hereafter Argo SSS). Argo data is collected and made freely available by the International Argo Program and the national programs that contribute to it (http://www.argo.ucsd.edu,http://argo.jcommops. org). The Argo Program is part of the Global Ocean Observing System.



### 3.1.3 Reanalysis Sea Surface Salinity

We are also interested in analyzing the strengths and weaknesses of the satellite products when compared with a reanalysis
product. To this end and for completeness, we have used the SSS fields provided by ARMOR3D Near Real-Time (Nardelli,
2012; Nardelli et al., 2016; Droghei et al., 2016) corresponding to the year 2017. We use version the 4 of ARMOR3D which
is freely available in the Copernicus Marine Environment Monitoring Service (CMEMS) service desk (https://resources.marine.
copernicus.eu/?option=com_csw&task=results?option=com_csw&view=details&product_id=MULTIOBS_GLO_PHY_REP_
015_002). In the generation of the SSS fields provided by ARMOR3D (hereafter CMEMS SSS), a correction based on the
ISAS-CORA SSS field is applied as well as a combination of a Quality Control SSS measurements obtained from ISAS-
CORA (both distributed through CMEMS) and a high-pass filter of Reynolds SST L4 satellite observations.

### 3.1.4 Sea Surface Temperature

OSTIA SST is used as a reference to assess the spatial structure, geophysical consistency, and the effective resolution of the
SSS satellite products (see section 2.1.2 for the complete description).

## 3.2 Validation methods

### 3.2.1 Comparison with Argo

Assuming that Argo values represent a ground truth (that is, we neglect representative errors that are however significant) we
have used Argo SSS to assess the biases and the standard deviations of the errors of the different SSS products. To that goal,
we temporally and spatially collocate the Argo SSS with the SSS maps as follows: every map is compared with the Argo SSS
available during the same period (9 days in the case of BEC products) used in the generation of that map. We compare the Argo
SSS with the value of the SSS product corresponding to the cell where the Argo is located. Before computing the s match-ups
between Arago and SSS product, we apply the following quality control over the values of Argo SSS:

- The cut-off depth for Argo profiles is taken between 5 and 10 m.

- Profiles from BioArgo and those included in the greylist (i.e., floats which may have problems with one or more sensors)
  are discarded.

- We use WOA2013 as an indicator: Argo float profiles with anomalies larger than 10°C in temperature or 5 psu in salinity
  when compared to WOA2013 are discarded.

- Only profiles having temperature close to surface between -2.5 and 40$^o$C and salinity between 2 and 41 psu are used.

### 3.2.2 Singularity analysis

Singularity analysis can be used for the assessment of the geophysical consistency among different products (Umbert et al.,
2014; Hoareau et al., 2018b). The singularity exponent (SE) at a given point is a measure of the regularity or irregularity of





a variable around that point (Turiel et al., 2005, 2008a, 2009). From an oceanographic point of view, SE are related to the advection term, and therefore, they are intrinsic characteristics of the flow and not specific to the chosen scalar (Nieves et al., 2007). The singularity fronts (bright white streamlines in Figure A11) clearly correspond to the general circulation features,

such as the Gulf Stream, the Kuroshio Extension and the tropical instability waves in both the Pacific and Atlantic, among others. This is evident in the singularity fronts derived from OSTIA SST shown in Figure A11. As discussed in Umbert et al. (2014), the SE of different scalars must be in correspondence, and as OSTIA SE have the better quality, we take them as a reference: the SE of all the SSS products will be compared to this reference. Since the OSTIA SST product has the highest resolution ($0.05 \times 0.05°$), prior to compute its SE it is first regridded at the same resolution of the SSS product it is going to

be compared with. In the case of L3 satellite products and CMEMS, this implies reducing the resolution to $0.25 \times 0.25°$ and 9-day. In the case of BEC L4, since it has the same spatio-temporal grid as OSTIA SST, no regridding is required prior to the calculation of SE.

A way to assess the correspondence of SE is to calculate conditioned histograms of one product SSS SE, $h_{SSS}$, by the values of OSTIA SE, $h_{SST}$. Let us recall the definition of the histogram of one random variable $y$ conditioned by one random variable

$x$, denoted by $\rho(y|x)$ and defined by the following expression:

$$\rho(y|x) \equiv \frac{\rho(x,y)}{\rho(x)} \tag{6}$$

where $\rho(x,y)$ is the joint histogram of both variables (the standard 2D histogram) and $\rho(x)$ is the marginal histogram of the variable $x$ (the standard histogram of this variable). In essence, if we put the variable $x$ in the abscissa axis and the variable $y$ in the ordinate axis, the conditioned histogram corresponds to taking the joint histogram of $x$ and $y$ and normalizing it by

columns so each column sums up 1.

The histogram of a variable conditioned by the value of another variable serves to evidence any functional dependence between both. Conditioned histograms have been used to put in evidence the correspondence of singularity exponents of two variables (Hoareau et al., 2018b), and we will use them for the same purpose here, with $h_{SST}$ the conditioning variable and $h_{SSS}$ the conditioned variable. When the modal line, defined by the maximum probability of $h_{SSS}$ for each value of

$h_{SST}$, is close to a straight line of slope 1 (the identity function), then the singularity exponents of the SSS maps are in good correspondence with those of the reference (Turiel et al., 2008a; Olmedo et al., 2016). In practice, this line becomes horizontal beyond a certain threshold value due to the increase of the error in the estimation of SE for larger values of SE (Turiel et al., 2006), as $h_{SSS}$ and $h_{SST}$ become independent of each other because they are dominated by noise. Notice that when two variables are independent, the conditioned histogram is constant in $x$ and thus the modal line becomes horizontal. In addition

to becoming horizontal, the conditioned dispersion will also be larger beyond that threshold, since any dependence between the variables would decrease the dispersion. Thus, besides analyzing the range of the singularity correspondence, the conditioned dispersion (i.e the standard deviation of the $h_{SSS}$ for each bin of $h_{SST}$) must also be considered.

In the following, we detail the methodology employed in this work:





- **Calculation of the histograms:** Regarding the calculation of the histograms shown in this paper, for a given SSS product we have accumulated all the values of the pairs ($h_{SST}$, $h_{SSS}$) of all the globe and for the whole year 2017 to create the joint histogram. Notice that, as shown in Umbert et al. (2014), the histograms of SE do not change in space nor in time and thus this accumulation can be done. The values SE used to construct the histograms have been limited to the range at [-0.4, 0.4] (that typically contains more than 80% of all possible values) and we fix the bin size to 0.05 for both $h_{SST}$ and $h_{SSS}$. The marginal histogram $\rho(h_{SST}$ is obtained by summing the values at each column; and the conditioned histogram is obtained by a simple division of all the elements of a column by the associated value of the marginal histogram.

- **Statistical descriptors:** We have computed three descriptors to characterize the conditioned histograms:

  - The most probable value of $h_{SSS}$ at each bin of $h_{SST}$: $H_0(h_{SST})$.

  - The mean value of $h_{SSS}$ at each bin of $h_{SST}$: $\bar{H}(h_{SST})$.

  - The standard deviation of $h_{SSS}$ at each bin of $h_{SSS}$: $\sigma_H(h_{SST})$.

- **Correspondence of the SE**: As a function of $h_{SST}$, both $H_0$ and $\bar{H}$ should ideally be close to the identitiy, what would be the best correspondence between $h_{SSS}$ and $h_{SST}$. Although the presence of physical processes other than flow advection, capable of creating singularities and affecting differently both variables (upwelling, rain-induced freshening or river plumes) could lead to a deviation from identity as the SE would not correspond, in general this effect is small and limited to very narrow areas, so the most frequent cause of deviation is just noise, that mainly affects the larger values of SE and for that reason the deviations show up as the value of SE increases. To quantify the quality of the correspondence between the SE of SST and those of SSS, we have computed the linear regression coefficients of the functions $H_0(h_{SST})$ and $\bar{H}(h_{SST})$ up to a given threshold value of $h_{SST}$, $h_{max}$. We then have compared the corresponding linear regression coefficients, $a_{mode}$ and $a_{mean}$ with 1: the closest $a_{mode}$ and $a_{mean}$ are to 1, the best the geophysical correspondence between SSS and SST in the range of $h_{SST} < h_{max}$.

- **Uncertainty analysis**: $\sigma_H(h_{SST})$ indicates which is the uncertainty associated to the correspondence between the SE of SST and SSS, which is due to effects that we cannot control (including noise, numerical inaccuracies of the algorithm used to compute SE and unknown physical processes). We have computed the mean value of $\sigma_H(h_{SST})$ in the same range of values of $h_{SST}$ used compute $a_{mean}$ and $a_{mode}$, namely $h_{SST} < h_{max}$. The goal is to have a value of mean $\sigma(h_{SSST})$ as small as possible compared to the marginal standard deviation of $h_{SST}$.

- **Validity range**: Finally, we have also analyzed which is the range of good correspondence between $h_{SST}$ and $h_{SST}$. The larger the range, the better, as this would ensure that more and more geophysical structures (fronts, eddies, filaments) are described by both variables



### 3.2.3 Spectral analysis

Spectral analysis has been extensively used to analyze satellite observations, in situ data and models outputs, both in the atmosphere and the ocean (Stammer, 1998; Reynolds and Chelton, 2010; Kolodziejczyk et al., 2015; Olmedo et al., 2016; Hoareau et al., 2018b). It is well-known that the Power Density Spectra (PDS) of a scalar submitted to turbulence should follow a power-lay behaviour, characterized by a scaling exponent, sometimes referred to as "spectral slope" as it corresponds to the slope of the log-log representation of the PDS as function of the wavenumber. The analysis of spectral slopes allows

obtaining information about the effective spatial resolution of the remote sensing data. For instance, the presence of noise makes the straight curve of log PDS vs log wavenumber to bend and become horizontal at high wavenumbers; this happens because noise is independent of the wavenumber but the amplitude of the signal decreases with the wavenumber, so at a given wavenumber large enough (and thus, at a given spatial scale small enough) noise is dominant: the crossover point signals the effective resolution of the data. Another situation that can appear is when the data is oversmoothened and then there is a

systematic lack of energy at high wavenumbers; in those case. a faster-than-linear decay is observed for wavenumber larger than the resolution threshold.

Theoretical studies predicted that temperature and salinity should have the same spectral slopes (Blumen, 1978; Charney, 1971). We have used the spectral analyses based on the PDS and on the Singularity Power Spectra (SPS), that corresponds to the PDS of SE maps (Hoareau et al., 2018b), to estimate the effective spatial resolutions of the SSS products. For this, we

compare the spectral slopes of the SSS products with the ones computed from OSTIA SST. PDS slopes are expected to be in between -1 and -3 depending on the dynamical regime that drives the ocean, while the expected SPS slopes for SST and SSS maps range between -2 and -2.5 (Hoareau et al., 2018b). Note that, while the PDS slope is affected or distorted by the presence of noise into the data, the SPS slope is not because the SE algorithm we use (Pont et al., 2013) is designed to filter noise. For the same reason, the use of SPS reduces the uncertainty in the determination of the spectral slopes (Hoareau et al., 2018b).

Therefore, both spectral methods are complementary from a validation point of view as PDS analysis gives information on the effective spatial resolution of the data, and SPS method assesses the existing geophysical structures beyond the remaining sources of uncertainty in remote sensing products.

The spectral analysis approach that we have followed in this work is the following one:

– We perform the spectral analysis over the same ocean regions proposed in Hoareau et al. (2018b) (see Figure A4).

– At each on of these regions, we compute the PDS from each SSS product and the OSTIA SST, as well as their corresponding SPS from their SE maps. Both spectra (PDS and SPS) are given as a function of wavenumber values in degree$^{-1}$ (latitude degrees for meridional regions and longitude degrees for zonal regions) and as wavelength values in kilometers. Recall that a wavelength contains a full oscillation from 0 to +1, then to -1 and finally back to 0, and therefore it contains two resolved points; therefore, to convert wavelengths to resolved spatial scales (resolution scale) the values

must be divided by 2.





- For each region and product, we have computed the mean PDS and the mean SPS over the full year 2017 to reduce the fluctuations of each individual spectrum.

- We have computed and analysed the slope of the averaged PDS and SPS, and we have compare them with the ones resulting from OSTIA SST.

### 390  3.2.4  Triple collocation

The triple collocation (TC) technique is a powerful tool to estimate the standard deviation of errors of three spatio-temporally collocated measurements of the same target. TC has been used to assess the quality of many remotely sensed variables, and in particular SSS (Hoareau et al., 2018a). The major assumptions of TC are that errors must be uncorrelated with the target variable and also that the errors of the different data sets must be uncorrelated between them. Some refined formulations have
been developed in recent years for taking into account the presence of cross-correlated errors between two of the data sets, but they require at least four data sets (Gruber et al., 2016; Pierdicca et al., 2017).

We have used a recently developed formulation of the triple collocation method, the Correlated Triple Collocation (CTC), for the case of three data sets that resolve similar spatial scales from which two of them present correlated errors (González-Gambau, 2020). This TC can be particularly beneficial for the error characterization of variables for which getting measurement
systems with uncorrelated errors is challenging or not feasible, and it is particularly well suited to work with limited samples of data because it has a fast convergence with the sampling size. This formulation has been proved for the characterization of radiometric errors in L-band brightness temperatures (TB). By the use of CTC, we have access to maps of errors, so we can characterize which places are less noisy, and also we can ascertain which is the best suited product depending on the location.

The triplets used in this analysis are shown in Table A1, indicating which of the three products is considered with uncorre-
lated errors with respect to the other two data sets. Errors among the different SMOS products are assumed to be correlated, as well as errors among the different SMAP products since they are measured by the same sensor. Additionally, we have assumed that errors of CMEMS and SMOS CATDS are correlated, since the later is assimilated by the former.

In order to estimate the SSS error of each product, we average the estimated errors resulting from each one of the triplets where the product is considered. We also compute the standard deviation of the estimated error in the different triplets as a
metric of the uncertainty in the error estimation.

### 3.3  Validation Results

#### 3.3.1  Comparison with Argo SSS

In the first part of this subsection, we present the comparison of the 9-year time series of the BEC SMOS SSS products with Argo SSS; then, in the second part, we extend the comparison to the rest of SSS products (see section 3.1.1) but for the year
2017 only. In Table A2, the regions of study analyzed in this section are presented.





Table A3 summarizes the statistics of the differences of BEC SMOS SSS v2.0 products minus Argo SSS for the 2011-2019 period in two different ocean regions : (i) the whole temperate ocean (labelled as GLO in table A2) and (ii) the tropics (labelled TR in the same table). The statistics are provided on a yearly basis.

In terms of mean differences with respect to Argo (that would account for biases in the products, if we consider Argo to be

the ground truth), the three BEC products (HR, LR, and L4) provide very similar performances. The BEC SSS v2.0 products have a mean difference with respect to Argo below 0.02 and 0.06 psu, in GLO and in TRO regions, respectively. Those values are rather small and may be statistically non-significant.

Regarding the standard deviation of the differences with respect to Argo salinity, among the three salinity fields of the v2 product, BEC HR is the one with the largest standard deviation and L4 the one with the lowest (BEC LR is in between the

other two). This is expected since BEC HR is known to have some high spatial frequency noise, while BEC LR is a smoothed version of the BEC HR salinity with a radius of 50 km and therefore a reduction in the noise is expected. The fusion technique used in the generation of the L4 also leads to a reduction of the random noise of the salinity maps, and it is even better than a simple low-pass filtering as it preserves fine-scale structures. The standard deviations of the differences between BEC products and Argo salinity range between 0.34 and 0.26 psu in the case of L3 HR; 0.27 and 0.24 psu in the case of BEC LR; and 0.24

and 0.21 psu in the case of L4. There is a significant reduction of the standard deviation of the differences between SMOS and Argo since 2017, which is more significant in the case of the BEC HR product. This reduction is probably due to a change in the spatial scales of the auxiliary data provided by ESA and what are used in the retrieval. Since 2017, the ECMWF auxiliary data (see section 2.1) are provided at a resolution of 7.8 km, while previously they were provided at 25 km.

We have extended the comparison with Argo to the other SSS products, for the year 2017 only. Figure A5 shows the spatial

arrangement of the differences between each salinity product (BEC L4 not shown) and the corresponding collocated Argo SSS, averaged in $5° \times 5°$ latitude-longitude cells during the year 2017. In the open ocean, CATDS and REMSS are the two products that provide the lowest differences. CATDS uses In Situ Analysis System (ISAS) SSS (Gaillard et al., 2016; Kolodziejczyk et al., 2018) for the generation of the debiased L3 maps (Boutin et al., 2018). ISAS SSS is an interpolated product of Argo SSS, and this could explain the low differences of CATDS with respect to Argo SSS. JPL displays the largest (positive) differences

with respect to Argo SSS. Regarding the BEC products, the three products display similar differences with respect to Argo SSS. Significant positive differences ($\approx 0.2$ psu) are evidenced in the North Atlantic Ocean and also in the North Pacific. Similar differences are found in the CMEMS product (especially in the Northern Atlantic and the Eastern Northern Pacific).

We have also analyzed the spatial arrangement of the standard deviations of the differences between the gridded products and Argo SSS in $5° \times 5°$ cells. Figure A6 shows these standard deviations for the different products (BEC L4 not shown). CMEMS

SSS presents the lowest standard deviation in some regions, such as the Southern Ocean and the Western North Atlantic. BEC HR is the product with the largest standard deviation. The rest of the satellite products show similar standard deviations. In the case of satellite products, the largest standard deviations are located in regions of strong salinity gradients as the Gulf Stream and close to the mouth of the main rivers. Discrepancies appear between the satellite products in the Southern Ocean, where SMOS products show a lower standard deviation of the difference than SMAP products.





Figure A7 shows the temporal evolution of the mean difference of the gridded products with respect to Argo SSS in $0.25°$
bands of latitude (BEC L4 not shown). The product with the lowest differences with respect to Argo SSS is CMEMS, probably
because CMEMS assimilate Argo data. Among the satellite products, BEC (HR, LR and L4) and REMSS present the lowest
latitudinal differences with respect to Argo. JPL shows the largest differences with respect to Argo SSS (increasing at high
latitudes).

We have calculated the statistics of the comparison with Argo SSS in the regions defined in table A2. Table A4 comprises
the results of the comparison for all the gridded products and Figure A8 summarizes the statistics by showing the mean (blue
square), standard deviation (purple bar), and root mean square difference (RMS, green circle) in all the defined regions. In the
plots, the labels correspond to each one of the analyzed products as follows: 1 corresponds to BEC HR; 2 to BEC LR; 3 to
CATDS; 4 to JPL; 5 to REMSS; 6 to CMEMS; and 7 to BEC L4. In general, BEC LR and BEC L4 provide very competitive
statistics with respect to Argo among all the satellite products. The only products that provide lower RMS differences with
respect to Argo in some regions are CMEMS with lower RMS in the Arctic Ocean and North Atlantic Ocean, and REMSS
with slightly lower RMS in Tropical and Equatorial regions, Equatorial Pacific, Amazon region and the Indian Ocean. The
processing of SMOS data in polar regions and also in semi-enclosed seas deserves specific algorithms, and in fact ESA is
funding several projects for developing dedicated products at the Mediterranean Sea, Baltic Sea, Arctic Ocean and Black Sea,
in which BEC is involved.

    We have also analyzed the temporal evolution of the difference with Argo statistics in Figures A9 and A10, for the mean and
standard deviation, respectively. The temporal evolution of the mean differences between the three BEC products and Argo are
very similar (see blue, green and orange lines in Figure A9) and stable, i.e., no significant oscillations are observed, specially
in GLO and TRO regions where the oscillations have an amplitude lower than 0.05 psu. The largest oscillations are observed in
the Arctic Ocean, where all the satellite products (except CMEMS) present annual variations larger than 0.4 psu, although not
all of these products evolve in the same way. In the case of the BEC products, the differences show a seasonal behaviour, being
negative in summer and positive in winter. In summer, fresh water masses coming from ice melting may remain in surface
because of stratification. This could produce negative differences between surface salinity (as measured by satellite SSS) and
a few-meter deeper salinity (Argo SSS). In winter, high winds may induce significant evaporation and also the formation of
sea ice produces excess salt, and this could explain part of the positive differences between measurements at surface and at the
first meters. Also notice that in the North Pacific, large differences appear at the end of the year. All the SMOS SSS products
present a negative difference with respect to Argo of -0.4 psu. REMSS also presents this negative difference at the beginning
but then it suddenly jumps to a positive difference.

    Regarding the temporal evolution of the standard deviation, there are some significant seasonal effects. For example, in the
northern hemisphere, Arctic and North Atlantic regions, the standard deviation is larger in winter than in spring-summer. This
is something expected because L-band TBs are less sensitive to the SSS in cold waters (winter time) than in warm waters
(summer time), what implies that the retrievals of SSS must be noisier in winter than in summer. However in the North Pacific,
all the satellite products present the inverse behaviour: a reduction of the standard deviation is present at the end of the year.
The reason of this decrease is still under study. In the Amazon river region, the standard deviation increases in spring and





summer times. The reason of this increase could be related to the seasonal behaviour of the North Brazil Current and the North Brazil Current Retroflection that has a seasonal behaviour which is manifested in the SSS (Castellanos et al., 2019), as well as to seasonal changes in the Amazon run off.

### 3.3.2 Singularity analysis

Figure A12 shows the histograms of the SE of each one of the SSS products conditioned by the SE of OSTIA SST. The modal
($H_0(h_{SST})$, in white) and the mean ($\bar{H}(h_{SST})$, in black) lines are also represented. In Figure A13 the statistical descriptors f (conditioned mean $\bar{H}(h_{SST})$, conditioned mode $H_0(h_{SST})$ and conditioned standard deviation $\sigma_H(h_{SST})$) of all the products are shown.

All the conditioned histogram present three well-defined different ranges. The first part of the curve is unstructured and noisy: this is normal because there are very few points with those values, so the statistics is scarce and fluctuations are large; it
is also affected by small mismatches in the positions of fronts, lack of accuracy of the sensors and, very occasionally, because of different singularity-inducing effects acting on each variable. Then, we find a central part where the relation between $h_{SST}$ and $h_{SSS}$ is clearly linear, as it is evident from the $H_0$ and $\bar{H}$ curves. After this, we find a third range, where the value of the conditioned histogram is saturated and the values of both $H_0$ and $\bar{H}$ are horizontal lines, indicating that $h_{SST}$ and $h_{SSS}$ are independent. This is also expected, as noise becomes dominant as we go to the largest values of the SE, and the noise in SSS
and in SST are independent. We can then separate the three ranges in the curve by two tipping points, one in the negative range (that we will denote by $h_{SST}^-$ and has a value typically around -0.3 or -0.2) and the other in the positive range (that we will denote by $h_{SST}+$ and has a value around 0.1). The most interesting range is the central one, which is delimited by these two tipping points, where the linear dependence between the SE of SSS and the SE of SST is observed; the larger is this central range (the geophysical correspondence range), the better.

All the products present uncorrelation between $h_{SSS}$ and OSTIA $h_{SST}$ from $h_{SST} > 0.1$. Therefore, we fix $h_{SST}^+ = 0.1$ for all SSS products.

We observe that the value of the $h_{SST}^-$ depends on the SSS product. Some of the products have good correlation from the most negative SE values while others start the correlated range in moderate negative SE values. For example BEC L4 (top right plot in Figure A12) presents a good correspondence between $h_{SSS}$ and $h_{SST}$ even at the most negative values of $h_{SST}$.
However, CMEMS product is far from the identity in the most negative values of the $h_{SST}$, and only at moderated negative values (from $h_{SST} \geq -0.2$) we see that the $h_{SST}$-$h_{SST}$ correspondence becomes closer to the identity.

**Quality of the geophysical correspondence:** We have computed $a_{mean}$ and $a_{mode}$ (and also $\langle \sigma_H \rangle$) in three intervals of $h_{SST}$: $[-0.4, 0.1]$, $[-0.3, 0.1]$ and iii) $[-0.2, 0.1]$. Table A5 shows the values of $a_{mean}$ and $a_{mode}$. We observe some differences between $a_{mean}$ and $a_{mode}$. This is because of $\bar{H}$ is a numerically more robust metric (it does not depend on the histogram
discretization), but it is more affected by outliers. On the contrary $H_0$ is numerically less stable (it depends on the histogram discretization) but it is less affected by outliers. In general, despite of their differences, both metrics are consistent when we intercompare them among the different SSS products. The only exception is REMSS that present a much lower $a_{mean}$ than $a_{mode}$. BEC L4 presents the best performance in all the intervals, having $a_{mean} > 0.68$ and $a_{mode} > 0.88$. The CMEMS





product presents also very high values of $a_{mean}$ and $a_{mode}$ but only in the interval of $[-0.2, 0.1]$; as mentioned above, in the

CMEMS product, the relation between $h_{SSS}$ and $h_{SST}$ in the most negative range are far from the identity. All the satellite SSS products provide negative values of $\bar{H}$ and $H_0$ for $h_{SST} \in [-0.35, -0.25]$, while CMEMS provides positive values. This suggest that there are some SSS fronts of moderate intensity that are captured by all the SSS satellite products but not captured at all by the CMEMS product. In the case of L3 satellite products, $a_{mean}$ and $a_{mode}$ range in between $0.29$ and $0.5$ depending on the product and the interval of analysis.

**Residual uncertainty:** For each of the ranges above, we have calculated the average on that range of values of $h_{SST}$ of the conditioned standard deviation, $\langle\sigma_h\rangle$- The best performance is for BEC L4 presenting a $\langle\sigma_h\rangle$ of 0.11 in all the analyzed ranges. The worse performances are for CMEMS that in the range of [-0.4, 0.1] reaches values of 0.17. The L3 satellite products provide similar $\langle\sigma_H\rangle$ that vary from 0.13 to 0.15 (depending on the product and the analyzed range).

**Extent of the geophysical correspondence range:** In order to determine the largest possible range of reasonable geophysi-

cal correspondence between SE of SSS and of SST, we have defined $h_{SST}^-$ as the most negative value of $h_{SST}$ with $a_{mean}$ (or $a_{mode}$) larger than 0.35. Table A5 shows $h_{sst}^+ - h_{sst}^-$ defined from $a_{mean}$ (11th column) and from $a_{mode}$ (12th column). The products that present the best performances are BEC L4, BEC LR and JPL.

### 3.3.3   Spectral analysis

Figures A14 and A15 represent the PDS and SPS (respectively) in the different regions for all SSS products and OSTIA SST

data. For a better comparison, Figure A16 presents the spectral slopes of the PDS (top) and SPS (bottom) for all the products together. These slopes are calculated in the range of 100-1000 km wavelengths. We observe a large diversity on the shapes of the SSS PDS (Figure A14) for the different SSS products, significantly differing of the OSTIA PDS (purple line). In contrast, the shapes of SSS SPS are closer to the shape of OSTIA SST SPS up to the 80km wavelength (see Figure A15). This is also observed in Figure A16: the values of PDS slopes vary on a range larger than the one of SPS slopes, which are more

concentrated around the theoretically expected range (between -2.0 and -2.5). These results indicate that, despite of the level of noise of each remote sensed product, the geophysical structures of the SSS data are consistent until a 100-80 km wavelength. However, the slope values in Figure A16 reveal some differences between the products:

- BEC HR (in blue) presents the flattest values of the PDS slopes being higher than -1.5 in most of the regions, indicating a strong influence of noise (noise tend to flatten the curve). However, the corresponding SPS slopes remain always in the

range of -2 and -2.5. This indicates that even if this product is the one with largest high-frequency noise, BEC HR allows describing consistently the geophysical structures up to 100km wavelength.

- CMEMS product (in grey) presents the steepest PDS slope, becoming lower than -3 in STP and SPURS regions, what means that the product is oversmoothened and that wide regions contain just plainly interpolated data. This indicates that CMEMS have a loss of structures at wavelengths larger than 100km. For example, Figure A14 shows that in STP

and SPURS regions, the slope of CMEMS (grey line) becomes steeper at wavelengths around 250km. This is partially





confirmed by the SPS slopes that in SPURS and NATL present values lower than -2.5, what indicates that fronts of SE have been lost and confirms the existence of an oversmoothing at wavelengths larger than 100km.

– CATDS (in green) presents the flattest PDS slope in the SPAC region ($\approx -1$). This suggests that the presence of noise in this region is very large in comparison with the other products and with its performance in other regions. SPAC region is used in the data processing of CATDS to correct systematic and temporal biases (it corresponds to the region where the OTT is computed and applied daily to the CATDS data (Tenerelli and Reul, 2010)). This result suggests that using this region for the calibration of the SMOS measurements could lead to some issues in the resulting product.

– In general, the L3 satellite products of BEC LR and CATDS have similar PDS slopes as well as JPL and REMSS in ITCZ, SPURS and STP regions. This should happen in all the regions, indicating that the spatial resolution of the products depends on the instrument, not on the methods used in the data processing. However, in ARC, NATL and SPAC, the PDS slopes of the BEC LR and CATDS and the ones of JPL and REMSS are significantly different between them, indicating the importance of the data processing methods to achieve proper spatial resolution.

– BEC L4 (in red) presents the closest PDS to those of OSTIA SST. Its SPS slopes remain in the range of -2 and -2.5 in all the regions. This indicates that BEC L4 allows describing spatial scales up to 100km wavelengths with the lowest presence of noise and the closest geophysical consistency with OSTIA SST data.

Below 100 km, except for BEC HR, all spectral slopes values get steeper (lack of signal variability into the data). REMSS (cyan line in Figure A14) presents a fast valley-shaped decay around wavelengths of 80km followed by a flattening (traduced by a steeper slope in the SPS slope). This indicates that the smoothing applied onto the REMSS product may remove part of the geophysical variability at those scales. Around 60km wavelength, BEC HR, BEC LR, CATDS and JPL PDS get flattened, while this does not happen in the corresponding SPS spectra. As SPS shape is less affected by noise (Hoareau et al., 2018b), these results indicate that despite of the noise, the geophysical signal present in BEC HR, BEC LR, CATDS, and JPL is consistently described even at those smaller scales, so they can be considered to be valid up to a wavelength of 80 km, which corresponds to a spatial resolution of 40 km.

### 3.3.4 Triple collocation

Figure A17 shows the estimated error standard deviations for the different SSS products (CMEMS not shown) for year 2017. As a general remark, the estimated error standard deviations are larger in those regions with higher salinity dynamics, as the Gulf of Bengal and the Equatorial Atlantic which is affected by the dynamics of the Amazon River plume. In general, BEC L4 provides the lowest error standard deviations among all the satellite products, with the exception of a region close to the Antarctic ice edge, where CATDS provides the lowest one. The uncertainty on the estimation of the error standard deviation (calculated as the standard deviation across all the possible triples of the error standard deviations) is lower than 0.03 psu for all the products and almost all ocean regions.





We assign one number to each product to assess which is the product with the lowest estimated error standard deviation at each ocean location. Figure A19 shows the four comparisons that have been performed:

- Comparison of BEC products: We as the label 1 to BEC HR, 2 to BEC LR and 3 to BEC L4. In general BEC L4 is the product with the lowest SSS error. However, BEC LR and BEC HR become more accurate in regions affected by regular rain events (such as the tropics) or continental fresh water discharges (such as Gulf of Mexico), where the hypothesis assumed in the generation of BEC L4 (the gradients of SSS and SST tend to be parallel) does not hold most of the year.

- Comparison of all satellite L3 products: We assign label 1 to BEC HR, 2 to BEC LR, 3 to CATDS, 4 to JPL and 5 to REMSS. In the bulk of the ocean BEC LR provides the lowest SSS error. In some specific regions, such as the Equatorial Atlantic and the Gulf of Guinea, which are regions strongly affected by the dynamics of the Amazon and Congo plumes, the BEC HR provides the best SSS error. Both SMAP products provide better SSS errors in regions affected by RFI (which is expected due to SMAP on-board RFI mitigation) such as the Chinese Sea, close to Madagascar and the Mediterranean Sea. In the Southern Ocean, CATDS is providing the best SSS error.

- Comparison of all satellite products: When we include BEC L4 in the comparison (1-BEC HR, 2-BEC LR, 3-BEC L4, 4-CATDS, 5-JPL and 6-REMSS), the smallest SSS error is given by BEC L4 in the majority of the ocean. As in the previous comparison, BEC HR and BEC LR provide the SSS with the lowest error in regions affected by rainfall and continental discharges. BEC L4 allows improving the SSS estimation in some regions affected by RFI with respect to the L3 products, such as in the China Sea and close to Madagascar, for example. In some regions, SMAP products provide the best SSS and in other BEC L4 is better.

- Comparison among all the products: In this case, BEC L4 remains the product with the lowest salinity error in most of the ocean regions. However, CMEMS SSS also appears to be the product with the lowest salinity error in many regions. For example, in the ocean regions close to Europe CMEMS SSS provides the best salinity estimation.

## 4 Conclusions

We have presented nine years of the new release of SMOS SSS global products generated at the Barcelona Expert Center: the BEC SMOS SSS global L3 and L4 products v2.0. The methods used in their generation include several improvements with respect to the previous version of these products: i) a new latitudinal-seasonal debiasing has been included; ii) an improved filtering criteria based on the salinity geophysical variability have been applied, which allows to better describe the salinity gradients without increasing the overall noise error in the maps; iii) new interpolation schemes are proposed to allow better describing small scale spatial features that are especially relevant in coastal regions; iv) the fusion scheme used in the generation of the L4 product has been modified to preserve small scale spatial features; v) an estimation of the salinity uncertainty is provided in the new products.

We have performed an extensive validation of the BEC SMOS SSS products v2.0. For doing this, we have compared the nine-year time series of the new BEC SMOS SSS with Argo uppermost salinity, and we have also compared the performance of





BEC products with other three satellite SSS products (the SMOS product produced at CATDS, two SMAP products generated
by REMSS and JPL, and the reanalysis product distributed by CMEMS, but in this case restricted to year 2017. The main
conclusions of this comparisons are:

– The statistics of the comparisons with Argo salinity evidence a competitive performance in comparison with the statistics
of the rest of SSS products. This includes small mean and standard deviation of the differences with respect to Argo SSS
(in the global and regional statistics, latitudinal biases and stable differences in terms of temporal evolution). In this
sense, the mean differences with respect to Argo SSS among the three BEC products (BEC L3 (HR and LR) and BEC
L4) are very similar (being lower than 0.02 psu at global scale), but the standard deviation is significantly different
among them, being the BEC HR the one with the largest standard deviation (being lower than 0.34 psu at global scale)
and BEC L4 the one with the lowest one (lower than 0.27 psu).

– In terms of effective spatial resolution and geophysical consistency, we have used two different metrics:

– Singularity analysis: the SE of BEC HR and BEC LR SSS products are very similar to the ones of the other satellite
salinity products in terms of correlation with OSTIA SST SE. A clear improvement is observed in the BEC L4 that
present a higher correlation with the SE of OSTIA SST, suggesting that the geophysical consistency is the most
accurate as it is the closest to OSTIA SST SE.

– Spectral analysis: The effective spatial resolutions of BEC HR and BEC LR are consistent with the ones of the
other satellite products which are at least a wavelength of 80 km (i.e., spatial resolution of 40 km). The BEC L4
presents similar spectral slopes to the ones of OSTIA SST, showing consistent slopes up to 50 km wavelength (25
km spatial resolution). At smaller scales, BEC L4 presents evidence of lack of structure and oversmoothening, so
probably it is not resolving scales at its nominal resolution of $0.05°$. In the case of BEC HR, it presents the flattest
PDS slopes but with SPS slope values between -2 and -2.5, which indicate that even if the presence of noise is
larger, BEC HR is able to represent consistently the geophysical signal.

– We have also computed an estimation of salinity errors by using triple collocation. Among the BEC products, BEC
L4 provides the SSS field with the lowest error, but in regions strongly affected by rainfall and continental freshwater
discharge, the L3 products (BEC HR and BEC LR) are better in terms of salinity error. When we compare all satellite
products, BEC L4 remains as the product with the overall minimum salinity error.

## 5   Data availability

The product is available on the website of the CMEMS Lambda project http://www.cmems-lambda.eu/ and in the Barcelona
Expert Center ftp service: http://bec.icm.csic.es/bec-ftp-service/. This product is also available trough EMODNET website
(https://www.emodnet-physics.eu/map/Products/Smos/). The doi of the level 3 product is http://dx.doi.org/10.20350/digitalCSIC/
12601 (Olmedo et al., 2020a) and the doi of the level 4 product is http://dx.doi.org/10.20350/digitalCSIC/12600 (Olmedo et al.,
2020b)





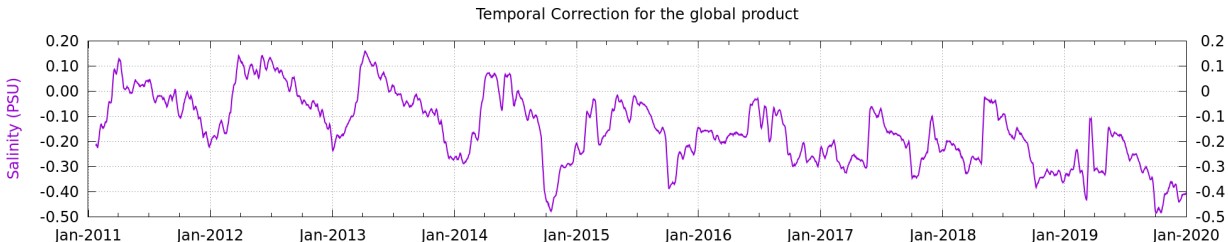

**Figure A1.** Temporal correction applied to the BEC SSS products.



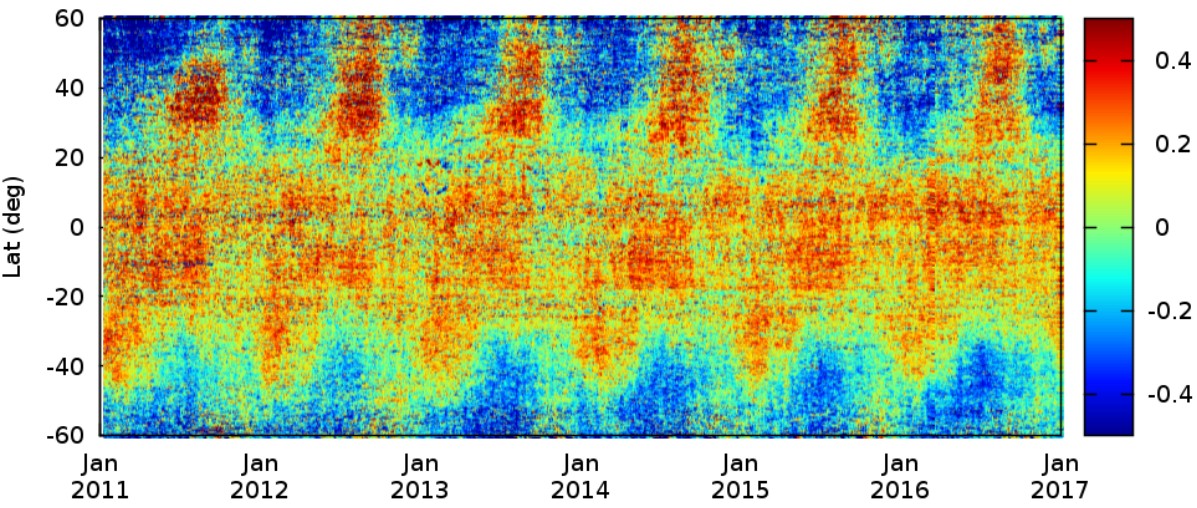

**Figure A2.** Latitudinal and seasonal bias affecting the BEC SMOS SSS L3 maps after applying the systematic bias correction proposed in section 2.2.1 and the temporal correction proposed in section 2.2.4. The plot represents a Hovmoller diagram of the temporal evolution of the differences between the BEC SMOS SSS and the Argo SSS in latitudinal bins of $0.25°$.

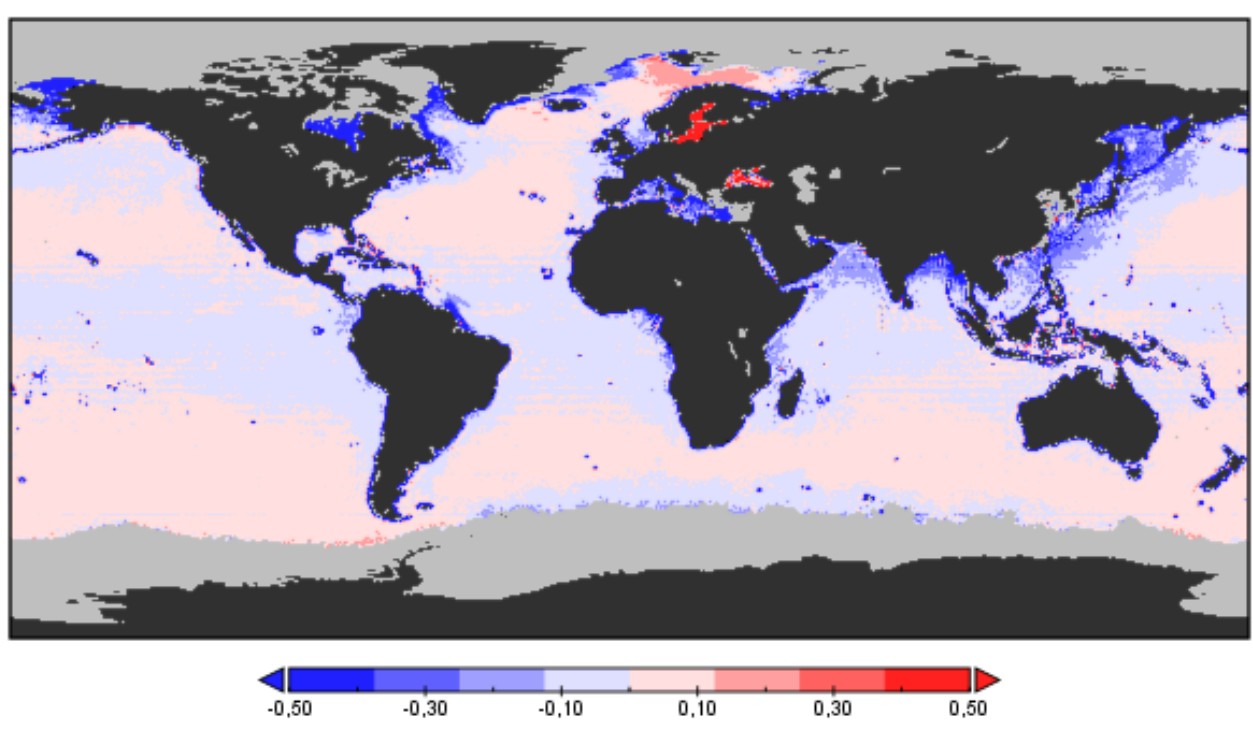

**Figure A3.** Difference between the 2011-2019 average of BEC SSS maps after applying corrections described in sections 2.2.1, 2.2.4 and 2.2.5 and WOA2013 SSS.

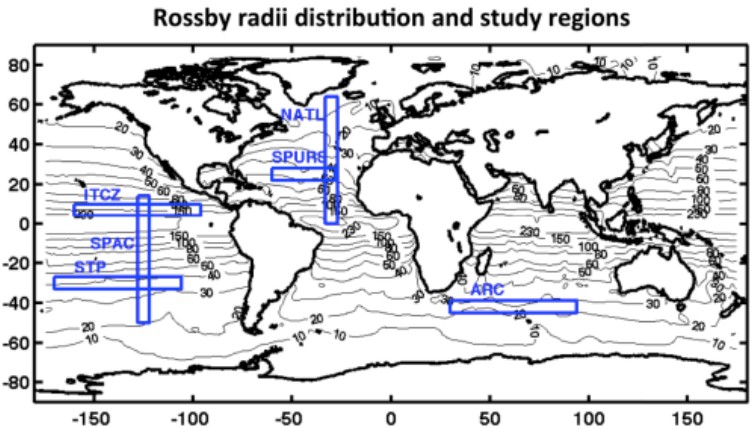

**Figure A4.** Spatial distribution of Rossby radii of deformation, from Hoareau et al. (2018b). Blue boxes are the regions where the Power Density Spectra and Singularity Power Spectra are computed: NATL [0-64°N, 27-33°W], ITCZ [4-10°N, 96-160°W], SPURS [22-28°N, 28-60°W], ARC [39-45°S, 30-94°E], STP [27-33°S, 106-170°W] and SPAC [14°N-50°S, 122-128°W].

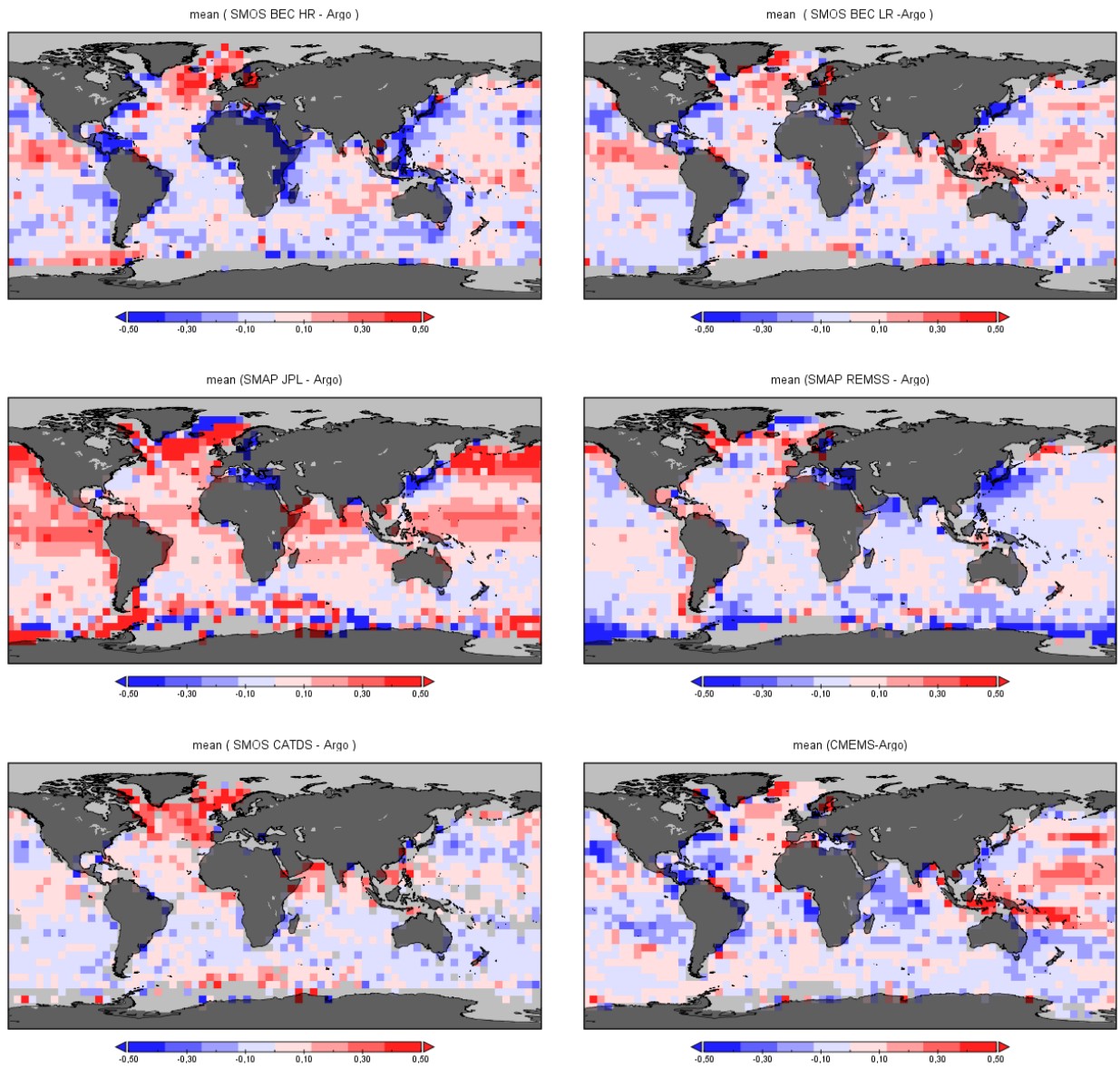

**Figure A5.** Spatial distribution of the mean differences with respect to Argo SSS. From left to right and top to bottom: BEC HR, BEC LR, JPL, REMSS, CATDS, CMEMS.



**Figure A6.** Spatial distribution of the standard deviations of the differences with respect to Argo SSS. From left to right and top to bottom: BEC HR, BEC LR, JPL, REMSS, CATDS, and CMEMS.

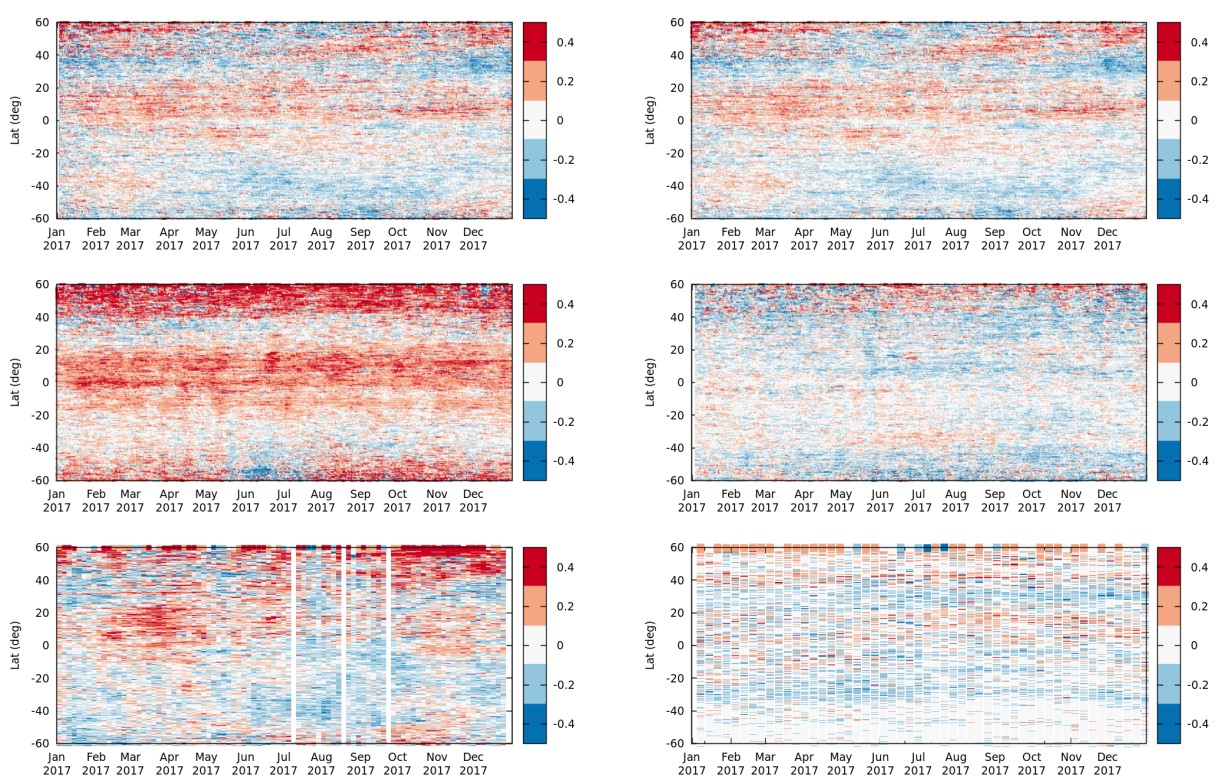

**Figure A7.** Hovmoller diagrams of the mean difference between salinity gridded products and the temporal and spatial collocated uppermost salinity measurement provided by Argo floats. The y-axis represents the latitudes: differences are averaged in latitude bins of 0.25 deg. The x-axis represents the time (in days): only the year 2017 is considered in this analysis. The gridded products used in this analysis are the following ones: (from left to right and top to bottom) BEC HR, BEC LR, JPL, REMSS, CATDS, and CMEMS.

**Figure A8.** Regional statistics of the comparison with respect to Argo salinity. The numbers in the label plots correspond to the different products that are compared with Argo SSS: 1-BEC HR; 2-BEC LR; 3-CATDS; 4-JPL; 5-REMSS; 6-CMEMS; and 7-BEC L4.



**Figure A9.** Mean of the differences between the gridded SSS products and Argo salinity in different ocean regions for the year 2017.



**Figure A10.** Standard deviation of the differences between the gridded SSS products and Argo salinity in different ocean regions for the year 2017.

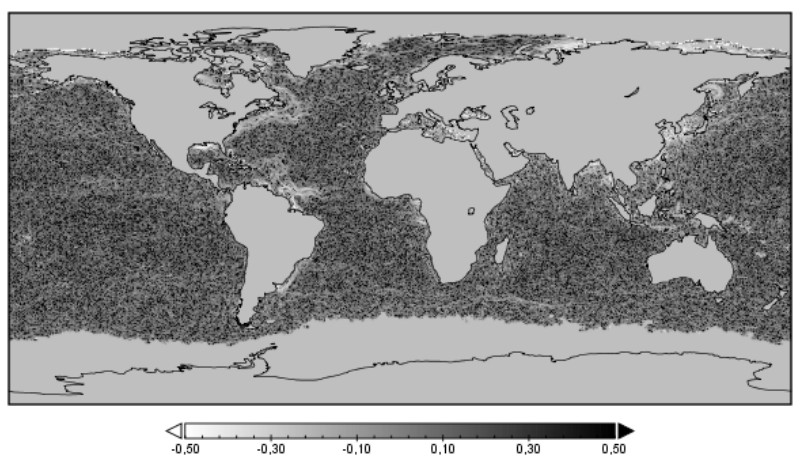

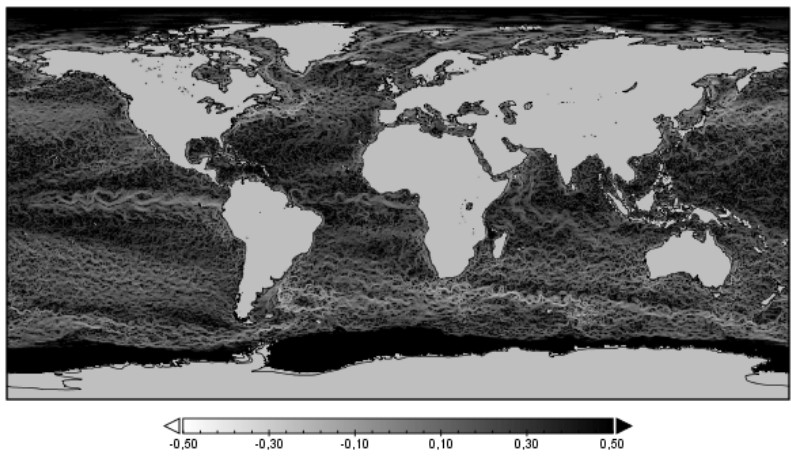

**Figure A11.** SE for the 15th August 2017. Top: SE of BEC LR SSS product. Bottom: SE of OSTIA SST.

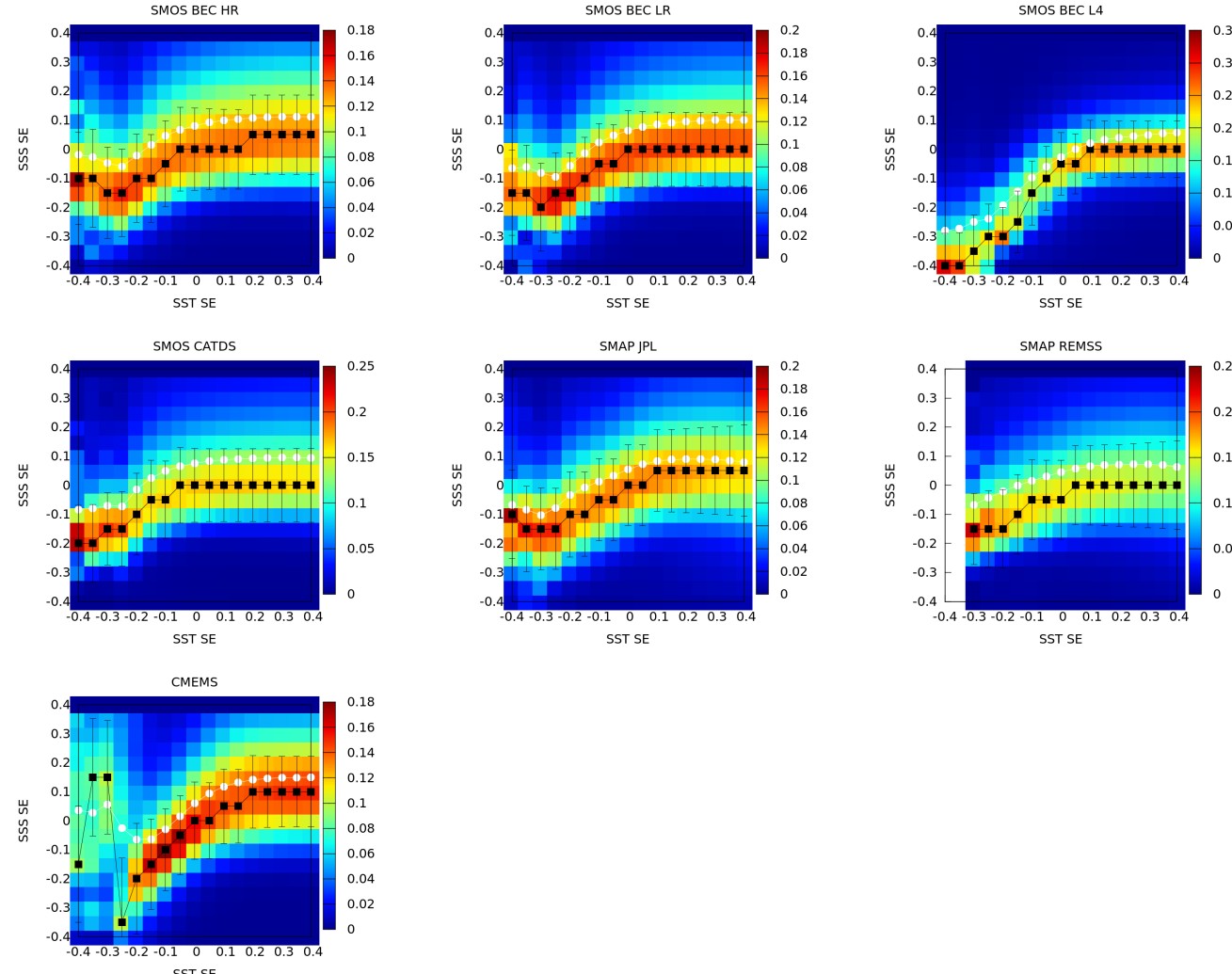

**Figure A12.** Histogram of SE of SSS conditioned to the SE of OSTIA SST. For each SST SE bin, the corresponding SSS SE distribution is normalized by the total number of SSS SE. The black line corresponds to the mode of the SE SSS at each bin of SE SST. The white line corresponds to the mean SE SSS ar each bin of SE SST. First row from left to right the SSS products correspond to BEC HR, BEC LR and BEC L4; Second row from left to right: CATDS, JPL and REMSS; and Third row CMEMS.





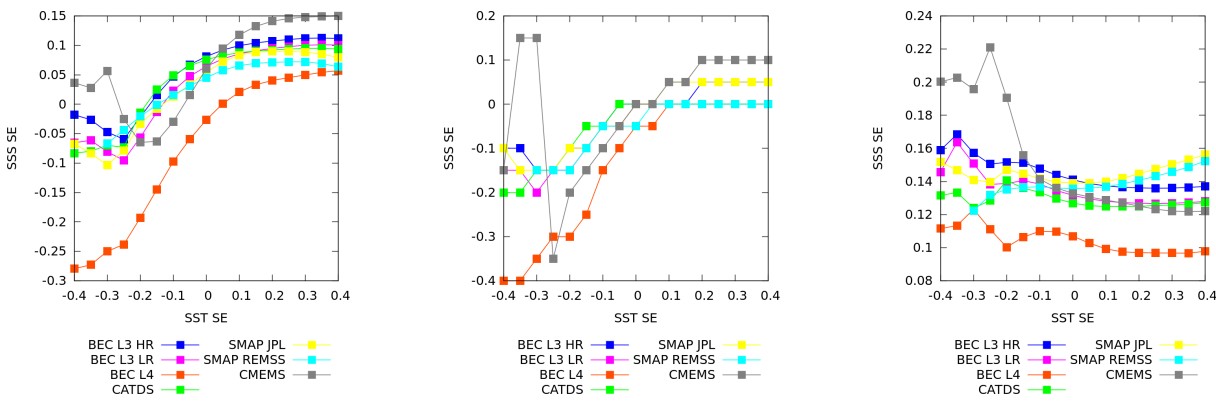

**Figure A13.** From left to right: The most probable SSS SE value as function of the SST SE; the mean SSS SE value as function of the SST SE; and the standard deviation of the SSS SE for each SST SE.



**Figure A14.** Power Density Spectra of the different SSS products.



**Figure A15.** Singularity Spectral Analysis of the different SSS products.



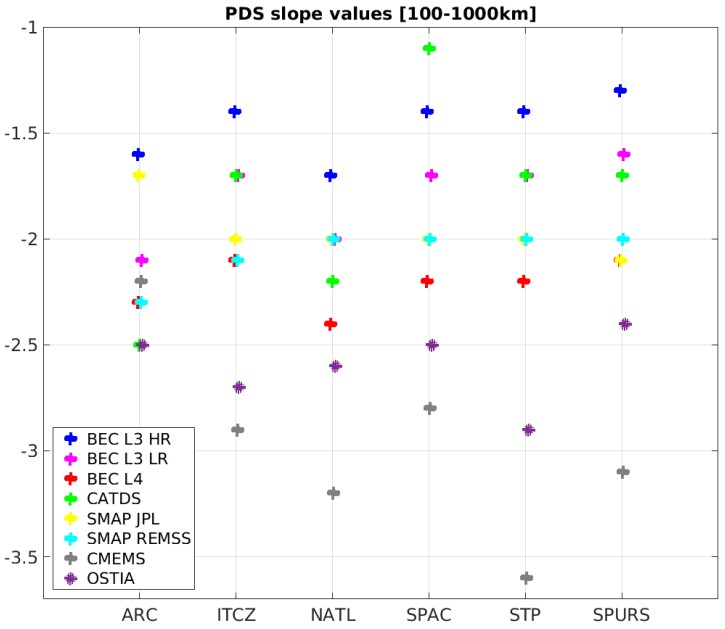

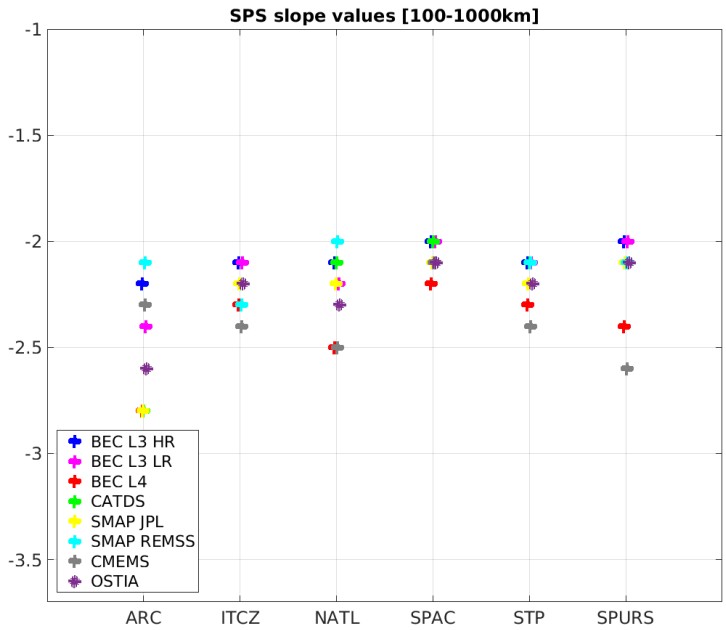

**Figure A16.** Slopes of the Power Density Spectra (top) and the Singularity Spectre Analysis (bottom) of the different SSS products. The corresponding slopes of the PDS and SPS of OSTIA SST are also included as a reference.



**Figure A17.** SSS error estimation by triple collocation, from top to bottom and left to right: BEC HR, BEC LR, BEC L4, JPL, REMSS, and CATDS.

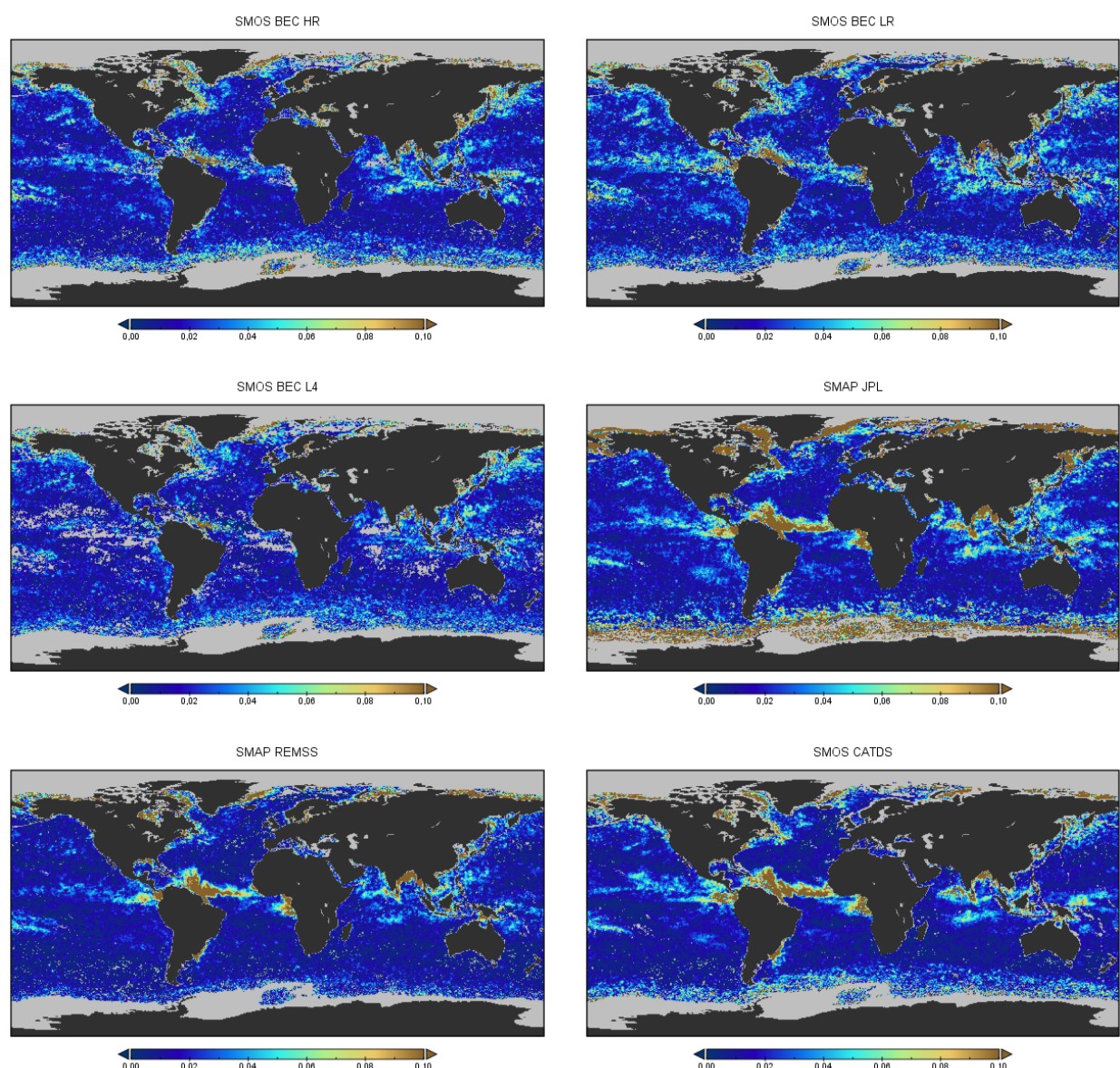

**Figure A18.** Uncertainty in the SSS error estimation by triple collocation, from, top to bottom and left to right: BEC HR, BEC LR, BEC L4, JPL, REMSS, and CATDS.

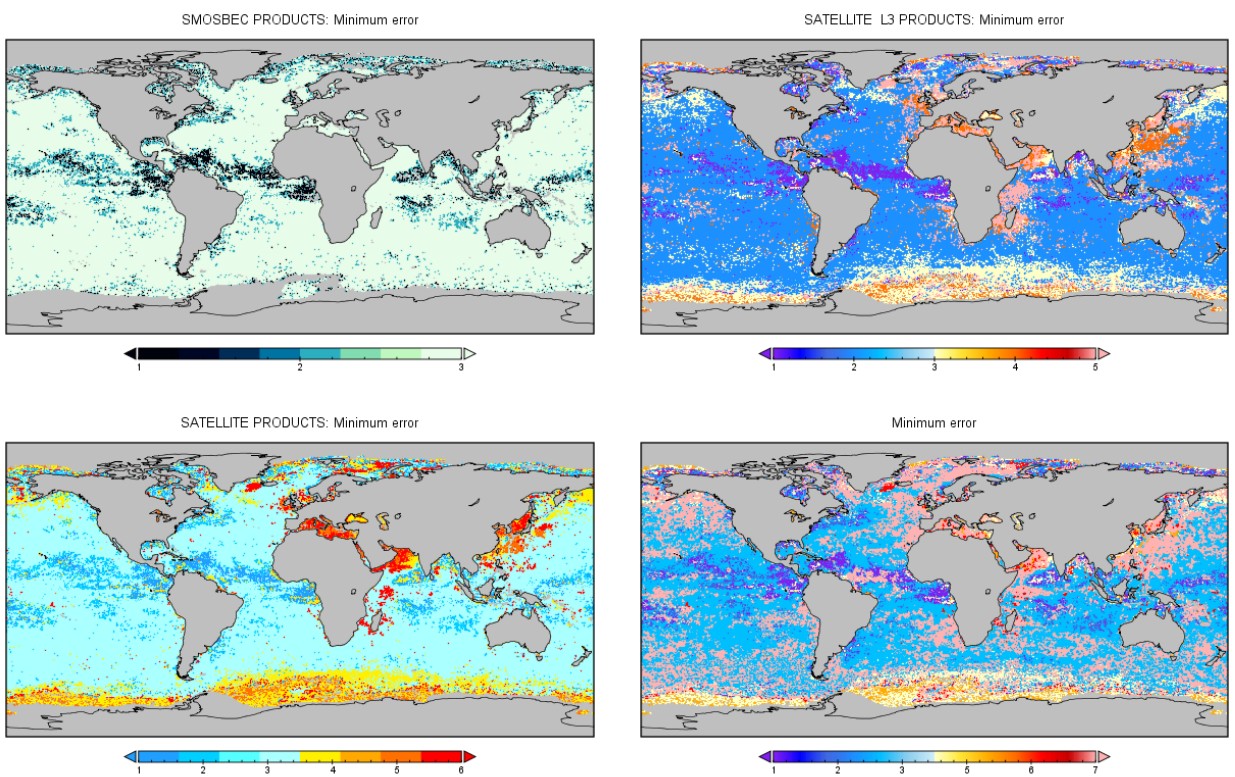

**Figure A19.** Spatial distribution of the products with the minimum SSS estimated error. Top left: The product with the lowest error among the BEC products (1-BEC HR; 2-BEC LR; and 3-BEC L4); Top right: The product with the lowest error among the L3 satellite products (1-BEC HR; 2-BEC LR; 3-CATDS; 4-JPL; and 5-REMSS); Bottom left: The product with the lowest error among the satellite products (1-BEC HR; 2-BEC LR; 3-BEC L4; 4-CATDS; 5-JPL; and 6-REMSS); and bottom right: the product with the lowest error among all the products analyzed in the study (including reanalysis) (1-BEC HR; 2-BEC LR; 3-BEC L4; 4-CATDS; 5-JPL; and 6-REMSS; 7-CMEMS).





**Table A1.** SSS triplets used in the analysis performed in section 3.3.4.

| Correlated source 1 | Correlated source 2 | Uncorrelated source |
|---|---|---|
| BEC HR | CATDS | JPL |
| BEC HR | CATDS | REMSS |
| BEC HR | CMEMS | JPL |
| BEC HR | CMEMS | REMSS |
| REMSS | JPL | BEC HR |
| BEC LR | CATDS | JPL |
| BEC LR | CATDS | REMSS |
| BEC LR | CMEMS | JPL |
| BEC LR | CMEMS | REMSS |
| REMSS | JPL | BEC LR |
| BEC L4 | CATDS | JPL |
| BEC L4 | CATDS | REMSS |
| BEC L4 | CMEMS | JPL |
| BEC L4 | CMEMS | REMSS |
| REMSS | JPL | BEC L4 |
| REMSS | JPL | CATDS |
| REMSS | JPL | CMEMS |



**Table A2.** Ocean regions used in the statistics with respect to Argo SSS

| Region | Description | Latitude | Longitude |
|--------|-------------|----------|-----------|
| GLO | Tropics and mid-latitudes | 60° S - 60° N | All |
| TRO | Tropics | 30° S - 30° N | All |
| EQU | Equatorial regions | 10° S - 10° N | All |
| ANT | Antarctic | 90° S - 50° S | All |
| ARC | Arctic | 50° N - 90° N | All |
| SPA | Southern Pacific | 30° S- 0° S | 150° W - 120°W |
| NAT | North Atlantic | 30° N- 50° N | 50° W - 0° W |
| AMA | Amazon river region | 0°N -20°N | 70○ W - 40° W |
| EPA | Equatorial Pacific | 10° S - 10° N | 180° W - 80° W |
| NPA | North Pacific | 30° N - 50° N | 180° W - 120° W |
| SAT | Southern Atlantic | 40° S - 0° S | 30° W - 0° W |
| IND | Indic Ocean | 30° S - 0° S | 60° E - 120° E |





| Year | Product | SMOS - ARGO in GLO | | | SMOS - ARGO in TRO | | |
|------|---------|--------|-------|-------|--------|-------|-------|
| | | \<Mean\> | \<STD\> | \<RMS\> | \<Mean\> | \<STD\> | \<RMS\> |
| 2011 | BEC HR | -0.02 | 0.34 | 0.34 | -0.03 | 0.31 | 0.31 |
| | BEC LR | -0.02 | 0.27 | 0.27 | -0.03 | 0.25 | 0.25 |
| | BEC L4 | -0.02 | 0.22 | 0.22 | -0.04 | 0.21 | 0.21 |
| 2012 | BEC HR | -0.02 | 0.34 | 0.34 | -0.04 | 0.31 | 0.31 |
| | BEC LR | -0.02 | 0.27 | 0.27 | -0.04 | 0.25 | 0.26 |
| | BEC L4 | -0.02 | 0.22 | 0.23 | -0.05 | 0.21 | 0.22 |
| 2013 | BEC HR | -0.03 | 0.34 | 0.34 | -0.03 | 0.33 | 0.33 |
| | BEC LR | -0.02 | 0.27 | 0.27 | -0.03 | 0.27 | 0.27 |
| | BEC L4 | -0.03 | 0.23 | 0.23 | -0.04 | 0.22 | 0.23 |
| 2014 | BEC HR | -0.01 | 0.34 | 0.34 | -0.02 | 0.32 | 0.32 |
| | BEC LR | -0.01 | 0.27 | 0.27 | -0.02 | 0.27 | 0.27 |
| | BEC L4 | -0.02 | 0.23 | 0.23 | -0.03 | 0.23 | 0.23 |
| 2015 | BEC HR | -0.00 | 0.34 | 0.34 | 0.00 | 0.32 | 0.33 |
| | BEC LR | -0.00 | 0.28 | 0.28 | 0.00 | 0.27 | 0.27 |
| | BEC L4 | -0.01 | 0.23 | 0.23 | -0.01 | 0.23 | 0.23 |
| 2016 | BEC HR | -0.01 | 0.34 | 0.34 | 0.01 | 0.32 | 0.33 |
| | BEC LR | -0.01 | 0.28 | 0.28 | 0.01 | 0.27 | 0.28 |
| | BEC L4 | -0.02 | 0.24 | 0.24 | 0.00 | 0.24 | 0.24 |
| 2017 | BEC HR | 0.00 | 0.28 | 0.28 | 0.03 | 0.27 | 0.27 |
| | BEC LR | -0.00 | 0.24 | 0.24 | 0.03 | 0.24 | 0.24 |
| | BEC L4 | -0.00 | 0.22 | 0.22 | 0.02 | 0.22 | 0.22 |
| 2018 | BEC HR | 0.01 | 0.27 | 0.27 | 0.04 | 0.26 | 0.26 |
| | BEC LR | 0.01 | 0.24 | 0.24 | 0.04 | 0.23 | 0.23 |
| | BEC L4 | 0.01 | 0.21 | 0.21 | 0.03 | 0.21 | 0.21 |
| 2019 | BEC HR | 0.02 | 0.27 | 0.27 | 0.06 | 0.25 | 0.26 |
| | BEC LR | 0.02 | 0.24 | 0.24 | 0.06 | 0.23 | 0.23 |
| | BEC L4 | 0.02 | 0.21 | 0.21 | 0.05 | 0.21 | 0.21 |

**Table A3.** Statistics of the comparison of the BEC global SSS product v2.0 with Argo



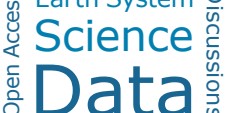

**Table A4.** Regional statistics of the differences between the gridded and Argo SSS for the year 2017. For each product and ocean region the mean, standard deviation and the root mean square of the difference are provided separated by :.

| Region | BEC HR | BEC LR | BEC L4 | CATDS | JPL | REMSS | CMEMS |
|--------|--------|--------|--------|-------|-----|-------|-------|
| GLO | 0.00:0.28:0.28 | -0.00:0.24:0.24 | 0.00:0.22:0.22 | 0.00:0.29:0.29 | 0.11:0.28:0.30 | -0.03:0.24:0.24 | -0.03:0.21:0.21 |
| TRO | 0.03:0.27:0.27 | 0.03:0.24:0.24 | 0.02:0.22:0.22 | 0.01:0.25:0.26 | 0.14:0.22:0.26 | -0.01:0.19:0.19 | -0.03:0.25:0.25 |
| EQU | 0.06:0.29:0.29 | 0.06:0.26:0.27 | 0.05:0.25:0.25 | 0.02:0.26:0.27 | 0.19:0.23:0.30 | -0.01:0.20:0.20 | -0.02:0.29:0.29 |
| ANT | -0.05:0.30:0.31 | -0.05:0.23:0.25 | - | -0.02:0.30:0.31 | 0.10:0.61:0.66 | -0.12:0.43:0.45 | -0.02:0.09:0.10 |
| ARC | 0.09:0.37:0.39 | 0.09:0.31:0.34 | 0.09:0.29:0.32 | 0.22:0.36:0.44 | 0.43:0.51:0.69 | 0.07:0.38:0.40 | 0.06:0.14:0.16 |
| SPA | -0.00:0.19:0.19 | -0.00:0.16:0.17 | -0.01:0.14:0.14 | -0.03:0.19:0.20 | 0.07:0.18:0.20 | 0.03:0.15:0.15 | -0.05:0.18:0.19 |
| NAT | -0.02:0.29:0.30 | -0.01:0.25:0.26 | -0.02:0.22:0.23 | 0.10:0.32:0.35 | 0.11:0.28:0.32 | 0.01:0.28:0.29 | 0.01:0.17:0.17 |
| AMA | 0.03:0.36:0.37 | 0.02:0.34:0.35 | 0.01:0.33:0.34 | 0.05:0.28:0.30 | 0.20:0.24:0.32 | -0.01:0.21:0.21 | -0.01:0.31:0.32 |
| EPA | 0.09:0.24:0.26 | 0.09:0.21:0.23 | 0.08:0.20:0.21 | 0.03:0.23:0.24 | 0.22:0.21:0.31 | 0.02:0.17:0.18 | -0.01:0.24:0.24 |
| NPA | -0.02:0.26:0.28 | -0.02:0.22:0.25 | -0.03:0.20:0.23 | -0.06:0.25:0.28 | 0.23:0.27:0.36 | -0.03:0.23:0.25 | 0.04:0.24:0.26 |
| SAT | -0.04:0.21:0.21 | -0.04:0.17:0.18 | -0.04:0.15:0.16 | -0.03:0.19:0.20 | 0.05:0.18:0.19 | 0.01:0.15:0.16 | -0.05:0.16:0.17 |
| IND | 0.03:0.23:0.23 | 0.03:0.20:0.21 | 0.03:0.18:0.19 | -0.03:0.21:0.22 | 0.11:0.19:0.23 | -0.01:0.16:0.16 | -0.08:0.22:0.24 |



**Table A5.** Singularity analysis metrics of the SSS products over three different SST SE regimes [-0.4, 0.1], [-0.3, 0.1] and [-0.2, 0.1]. For each one of these regimes we compute the linear regression coefficient of the mean SSS SE as function of SST SE (mean, $\bar{H}(h_{SST})$); the linear regression coefficient of the most probable value of SSS SE as function of the SST SE (mode, $H_0(h_{SST})$) and the averaged value of the standard deviation of the SSS SE as function of SST SE in the corresponding range ($\langle\sigma_H\rangle$).

| | Fitting range [-0.4, 0.1] | | | Fitting range [-0.3, 0.1] | | | Fitting range [-0.2, 0.1] | | | $h_{sst}^+ - h_{sst}^-$ | |
|---|---|---|---|---|---|---|---|---|---|---|---|
| | $a_{mean}$ | $a_{mode}$ | $\langle\sigma_H\rangle$ | $a_{mean}$ | $a_{mode}$ | $\langle\sigma_H\rangle$ | $a_{mean}$ | $a_{mode}$ | $\langle\sigma_H\rangle$ | from $a_{mean}$ | from $a_{mode}$ |
| BEC HR | 0.32 | 0.31 | 0.15 | 0.43 | 0.45 | 0.15 | 0.39 | 0.39 | 0.14 | 0.4 | 0.4 |
| BEC LR | 0.38 | 0.41 | 0.14 | 0.50 | 0.53 | 0.14 | 0.46 | 0.5 | 0.13 | 0.5 | 0.5 |
| BEC L4 | 0.68 | 0.88 | 0.11 | 0.74 | 0.93 | 0.11 | 0.72 | 0.99 | 0.11 | 0.5 | 0.5 |
| CATDS | 0.42 | 0.47 | 0.13 | 0.44 | 0.43 | 0.13 | 0.32 | 0.32 | 0.13 | 0.4 | 0.4 |
| REMSS | 0.33 | 0.43 | 0.13 | 0.33 | 0.43 | 0.13 | 0.29 | 0.46 | 0.14 | 0 | 0.5 |
| JPL | 0.39 | 0.37 | 0.14 | 0.47 | 0.50 | 0.14 | 0.40 | 0.50 | 0.14 | 0.5 | 0.5 |
| CMEMS | 0.15 | 0.12 | 0.17 | 0.31 | 0.38 | 0.16 | 0.68 | 0.82 | 0.15 | 0.3 | 0.4 |

*Author contributions.* E. Olmedo is the responsible of the development of the generation algorithms. She has generated the BEC product and is the main contributor to the writing of this manuscript. C. Gonzalez-Haro is the responsible of the distribution of the products. The validation of the products have been carried out by C. González, N. Hoareau, M. Umbert and E. Olmedo. V. Gonzalez-Gambau is the specialist of L1 data and calibration; she is also responsible of the triple collocation algorithm development and implementation. J. Martinez implemented all the algorithms for the correction of data at L2, as well as its georreferentiation. C. Gabarro is specialist in high-latitude salinity and contributed to the discussion of the issues at polar regions, jointly with J. Martinez. A. Turiel is the head of the BEC. He has participated in the development of all the algorithms (both product generation and validation). He supervised the whole manuscript, improving the mathematical and oceanographic description of several sections. All the coauthors have contributed in the writing and revision of the manuscript.

*Competing interests.* The authors declare that they have no conflict of interest.

*Acknowledgements.* This work has been carried out as part of the Copernicus Marine Environment Monitoring Service (CMEMS) Land-Marine Boundary Development and Analysis (Lambda) project. This work was funded by the Ministry of Economy and Competitiveness, Spain, through the National R+D Plan under L-Band Project ESP2017-89463-C3-1-R and previous grants and by the European Space Agency through the contract CCI+ Salinity. This work represents a contribution to CSIC Thematic Interdisciplinary Platform PTI Teledetect.





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
