# Peer review of "Nine years of SMOS Sea Surface Salinity global maps at the Barcelona Expert Center"

_Earth System Science Data, 2020_

## Referee Comment (RC1) · Anonymous Referee #1 · 30 Oct 2020

This paper describes a global sea surface salinity product produced by the Barcelona Center. It discusses the algorithms by which the SSS values are produced from raw brightness temperature. It then goes on to do a number of comparisons to Argo data and other SSS products, compute singularity exponents, display power spectra, etc.

The paper is thorough and appears complete. Thus it is very much worth publishing, largely as is. I have made a few comments of an editorial nature below. My only substantive argument with their methods is noted on line 169. It is not clear how they used the assumption of constant global average SSS, or whether it is even a very good assumption.

The authors need to go through all of the references to make sure they are correct and complete, including links. Many of them provide both a DOI link and one directly to

the publisher (e.g. Nieves et al). I would recommend deleting the direct links to the publisher, but that is a decision for the editors of ESSD. All references should include a DOI if available, or a URL for technical reports available online. Again, this should be according to the editorial policies of ESSD.

Line 62. The link given here may not lead to the correct place. It gets cut off at the line break. Ditto lines 79-80, 267-269, 772-773, etc. The authors need to check all links in this paper.

Lines 84-85. Is this the same SST as described above?

Lines 82-87. The references given here are not accessible, so I cannot check on the source of the ancillary data to see if it is properly described.

Lines 117-118. "...by subtracting each individual $s\_n\hat{}raw$ from the corresponding..."?

Line 138. Practical salinity is only defined in the range of [2 42]. See

Unesco (1981). The Practical Salinity Scale 1978 and the International Equation of State of Seawater 1980. Tech. Pap. Mar. Sci., 36.

Lines 144-145. These skewness and kurtosis criteria are not discussed. What is their purpose? Why the values given (1 and 2)?

Line 148. Where does the 25 come from?

Line 169. This is problematic. it is an assumption the authors are making, but it is not clear it is true. Cann they please provide a reference or some other justification.

Figure A1. It's interesting that the difference decreases over time. Can the authors interpret this?

Line 232. "spatial radio"?

Lines 291-292. Repeats from lines 205-210.

Line 329. Missing ")"

[Figure]

Line 358. "power-law" behaviour?

Line 434. Repeats from line 415. Delete.

Lines 456-459. "Figure A8..." This information is in the caption and does not need to be repeated.

Line 497. Not remembering the earlier section where this is described... $H_0$ is the black curve in Fig. A12 and H-bar is the white one? Put this in the caption.

Line 690. This reference is undecipherable. Provide a URL. Ditto the Sabater and de Rosnay reference.

Section 5. It was not obvious how to access the data from the emodnet or cmems sites. Visualizations were available, but not the data themselves.

---

## Referee Comment (RC2) · Anonymous Referee #2 · 28 Nov 2020

This paper presents a new reprocessing of the BEC SMOS SSS global L3 and L4 products v2.0 for a 9-years period comprising 2011 to 2019, with an updated methodology with respect to previous BEC version. The updates concern new filtering criteria, a new empirical correction for latitudinal and seasonal biases, an improved scheme for preserving small-scale gradients close to the coast, and the provision of a sea surface salinity uncertainty. Three BEC products are presented : a high resolution product, a low resolution product (smoothed with a radius of 50km) and a L4 product produced using a multifractal fusion technique applied to SMOS L3 using OSTIA SST fields.

After the methodology presentation, evaluation exercises are performed :

-by comparing the new BEC products with Argo salinities over the whole time series

-by comparing the caracteristics of the new BEC product relative to other satellite salinity products and to a CMEMS salinity product. These comparisons are restricted to the 2017 year. They concern not only comparisons to Argo but also comparisons of singularity exponents, of spectral properties and results obtained with triple colocations.

This is a huge piece of work by one of the two data centers delivering SMOS Level 3/4 products. Given the unique length of SMOS satellite salinity, it is a very important topic. The paper is clearly written and contains many interesting results. Nevertheless, I think several aspects would be worth to be deepened before publication, as this paper will serve as reference for future work:

- The 2017 year is chosen to intercompare the various products because it is close to a normal year ('there is not any large-scale geophysical phenomenon'). But given that the BEC products are calibrated using a climatology and that the methodology makes use of many filterings, it would be important to evaluate the product skills relative to others during a non normal year, e.g. 2016.

- The metrics used in the comparisons with Argo are mean, standard deviation and rms difference. They should also include R2 : indeed, the BEC methodology employs many filterings and R2 would allow to measure their efficiency in maintaining the SSS variability.

- By construction, the singularity exponents of the L4 product are expected to be close to the ones of OSTIA SST so I don't think that comparing BEC L4 and OSTIA SST properties provides an independent validation of BEC L4 products.

Detailed remarks :

Abstract : not sure doi are useful in the abstract.

Abstract : last statement (iii) should be moderated (see below my comments about the triple colocation section) : I don't think the authors have an absolute reference allowing a measure of the error. Given the huge spatial variability in regions affected by rainfall

and by continental freshwater discharges, I am surprised that a low resolution product is more accurate.

L20 : verb is missing

L126 : radiometric sensitivities wording is confusing (see Randa et al (Recommended Terminology for Microwave Radiometry, National Institute of Standards and Technology Technical Note 1551 (August 2008) : Radiometric sensitivity is often used to mean radiometric resolution, but this use is discouraged in view of the definition of sensitivity). I guess the authors mean radiometric resolution.

Section 2.2.3 : It is useful to recall all these filterings. It might be useful to point out more specifically the changes with respect to Olmedo et al 2017 and to display maps of the number of filtered points. In particular, the distribution of the salinity is naturally skewed in rainy regions and in river plumes, what is the effect of such filterings in these regions ?

Equation 4 corresponds to a one sigma sorting, which seems very stringent, what is the rationale for this choice ? how sensitive is the result to this ?

Figure A2 : I suggest to add a figure displaying the polynomial correction.

Section 2.2.6 : A major difficulty is how to filter SMOS retrieved salinity given that salinity distribution is very skewed and that RFI contamination might lead to artificial skewed distribution too. In the updated methodology, more stringent filterings than in Olmedo et al. (2017) are applied. After having performed the serie of filterings and corrections, an inconsistency between the WOA reference and the mean corrected field appears in river plumes which is likely an effect of the skewed salinity distribution (Figure A3) but it seems that there is also a global north-south difference : could the authors refine the color scale of Figure A3 to allow a better display (e.g. with a 0.02psu resolution)?

L243-247 : the (Boutin et al., 2016, 2018; Kolodziejczyk et al., 2016) publications refer to earlier versions of CATDS CEC products than version 4. A summary of the version 4 updates with respect to earlier versions is available on https://www.catds.fr/Products/Available-products-from-CEC-OS/CEC-Locean-L3-Debiased-v4.

Section 3.1.3 : It is indicated line 407 that CMEMS product assimilates SMOS CATDS SSS observations. This could be indicated in this section.

L294 : Figure A11 is mentionned in the text just after Figure A3. Figures might be reordered.

L351 : hsst mentionned twice.

Table A1 : CMEMS product

L415 : see my general comment about making the validation for the 'normal' year 2017. What would be obtained for year 2016 ?

L437-439 : I don't understand. CATDS CEC v4 fields use the combination of all ISAS fields over the 2012-2018 period to calibrate the full time series (see https://www.catds.fr/Products/Available-products-from-CEC-OS/CEC-Locean-L3-Debiased-v4: 'These relative salinity variations are then converted, in a last step, to salinities by adding a single constant determined, in each pixel, from SSS statistical distribution over the whole period (SMOS SSS distribution compared to ISAS SSS). This last step only determines the absolute SSS calibration in each grid point; the SMOS SSS temporal variation is independent of this adjustment.' In other words, both BEC and CATDS are calibrated using a climatology. CATDS does not use ISAS SSS to perform an adjustment to a specific year.

Figures A5 to A10 and Table 4: I recommend to add information about R2.

Figure A7 : Since the paper contains many figures, I suggest to remove Figure A7 which contains information in some way redundant with Figure A9. In case the authors prefer to keep Figure A7, it would be interesting, in addition to the longitudinal mean,

to display standard deviation.

Line 447 : 'In the case of satellite products,' could be removed.

Line 463 : Remove 'ESA is funding'.

Line 478 : This -0.4psu difference observed in December with all satellite products is surprising : is it homogeneously spread over the region or concentrated on a few grid points?

Section 3.3.2 : The singularity analysis is interesting as it is a way to check how natural gradients are retained by the various processings. However, I think the 'best performance' obtained with BEC L4 is artificial because BEC L4 products are built assuming that SE of OSTIA SST and of BEC L4 SSS should be similar. Hence the results obtained with BEC L4 on Figures A12 and A13 are not independent validation (section 3.3 is validation) but rather a verification of the proper behaviour of the L4 construction methodology (which is good to show). Hence the results presentation needs to be revised. L508 'For instance' might be replaced by 'By construction'. In the following, the performances obtained with BEC L4 should not be compared with the ones obtained with other products, but they might be presented as an 'ideal' case.

OSTIA SST is taken as a reference but OSTIA SST is not perfect. I guess SST SE could be different if derived with another SST field. This could be discussed a bit and maybe an order of magnitude of the uncertainty associated with OSTIA uncertainty could be derived?

Section 3.3.3 : what are the confidence levels on the spectra (Fig A14 & A15) and on their slopes ((Fig A16) ? In rainy regions like ITCZ can we really expect PDS of SST and SSS to be similar ?

L559-560 : I don't understand the meaning of this sentence: SMOS and SMAP are not the same instrument.

L563-565 : isn't it by construction ?

Section 3.3.4: I am puzzled with the results (or their interpretation) obtained with the triple collocation. According to Figure A17 and A19, BEC products would have the lowest errors in very variable SSS areas with respect to other products. Why ? Could it be an effect of filtering ? This result is surprising given the Argo comparisons presented in Table A4. For instance, from Table A4 in the Amazone region, the std difference and the RMS difference between satellite SSS and Argo salinity obtained with BEC products are the highest, which suggests a higher error with BEC products. This is opposite to what is indicated with Figure A17 and Figure A19. In the equatorial band, in Table A4, the REMSS products display the best comparisons to Argo which seems again opposite to the errrors displayed on Figures A17 and A19. The apparent contradiction between results obtained with triple colocation and Argo comparisons should be further explored before a conclusion on the error is drawn.

L584 : a verb is missing

L617-623 : a caveat for very variable regions (Amazone) should be added.

L626-628 : I expect this to be by construction (see previous comments)

On the figures displaying maps (A3, A5, A6, A17, A18, A19), latitude and longitude labels are missing.

Figure A13 : The y axis of last figure should be std(SSS).

Figure A13-A16 (in particular A14-A15) : colors are hard to see (especially yellow)

Table A2 : I guess Indic should be Indian ? (same remark for Figures A8-A10)

---

## Author Comment (AC1) · 30 Dec 2020

**Answers to Reviewer #1:**

This paper describes a global sea surface salinity product produced by the Barcelona Center. It discusses the algorithms by which the SSS values are produced from raw brightness temperature. It then goes on to do a number of comparisons to Argo data and other SSS products, compute singularity exponents, display power spectra, etc. The paper is thorough and appears complete. Thus it is very much worth publishing, largely as is. I have made a few comments of an editorial nature below.

Thank you very much for your comments.

My only substantive argument with their methods is noted on line 169. It is not clear how they used the assumption of constant global average SSS, or whether it is even a very good assumption.

We have checked this assumption by comparing the global average of a constant salinity reference (World Ocean Atlas 2013) against collocated Argo salinity measurements.
The following figure shows the difference between WOA 13 and Argo salinity measurements:

[Figure]

As observed in the figure, the assumption is valid up to hundredths of psu.
We have included this Panel in Figure 1 and we have modified the text accordingly.

The authors need to go through all of the references to make sure they are correct and complete, including links. Many of them provide both a DOI link and one directly to the publisher (e.g. Nieves et al). I would recommend deleting the direct links to the publisher, but that is a decision for the editors of ESSD. All references should include a DOI if available, or a URL for technical reports available online. Again, this should be according to the editorial policies of ESSD.

We have reviewed all the references. They, now, only contain their doi (or URL for technical reports) when they are available.

Line 62. The link given here may not lead to the correct place. It gets cut off at the line break. Ditto lines 79-80, 267-269, 772-773, etc.  The authors need to check all links in this paper.

We have edited the URL long links provided to make them more legible and make sure they lead to the correct place.

Lines 84-85. Is this the same SST as described above?

The auxiliary SST provided by ECMWF is based on SST OSTIA but it has been collocated in time and space with SMOS measurements.

Lines 82-87. The references given here are not accessible, so I cannot check on the source of the ancillary data to see if it is properly described.
The permissions to distribute this data are restricted to the Expert Support Laboratory teams and the private companies working on the mission. The European Space Agency (and also the European Center for Medium Range Weather Forecasts (ECMWF)) are the entities that manage this.
However this link:
https://smos-diss.eo.esa.int/oads/access/collection/AUX_Dynamic_Open
allows accessing to the data by searching SM_OPER_AUX_ECMWF* in the search box.

Lines 117-118. "...by subtracting each individual s_nˆraw from the corresponding..."?
We have modified the text as follows:
Lines 118-119:
.. *by subtracting the corresponding SMOS-based climatology $s^c(\gamma)$ from each individual $s_n^{raw}(\gamma)$….*

Line 138. Practical salinity is only defined in the range of [2 42]. See Unesco (1981). The Practical Salinity Scale 1978 and the International Equation of State of Seawater 1980. Tech. Pap. Mar. Sci., 36.
The reviewer is right. We have removed psu from that line. Although psu is defined in the range of [2:42],  due to the radiometric errors of the instrument, the retrieved salinity  from SMOS TBs could reach values that are out of this interval. Here, we extend the interval of "valid" salinity retrievals, because there are still some corrections in the methodology that have to be applied after this step that could lead to valid retrievals.

Lines 144-145. These skewness and kurtosis criteria are not discussed. What is their purpose? Why the values given (1 and 2)?
This is discussed in more detail in Olmedo et al 2017. The idea is that this approach is less accurate under non-Gaussian conditions. Skewness with absolute values larger than 1 are very skewed distributions. In this case, the definition of a

central estimator of the distribution, which is required for mitigating systematic biases, is less accurate. The same happens with the kurtosis. Kurtosis lower than 2 correspond to very flat distributions, where the definition of a central estimator is less accurate. We have added the following discussion in the text:

Lines 147-153:

*These filtering criteria are the same as the ones introduced in (Olmedo et al., 2017). The only difference is that now the criterion corresponding to the kurtosis is more relaxed: In (Olmedo et al., 2017) the set $\{s_n^{raw}(\gamma)\}$ was considered not valid and thus discarded out when the kurtosis of the distribution were larger than 4. Now we filter only platykurtotic distributions but not leptokurtotic ones. Regarding the impact of the filtering criterion corresponding to the skewness, this is the same as the one proposed in Olmedo et al. (2017). This criterion aims at discarding ocean regions affected by RFI contamination. Although some geophysical events tend to be not symmetric and fresh, as continental discharge and ice melting, and this leads to negative skewed salinity distributions, the typical skewness in these cases is around -0.5. The skewness values lower than -1 correspond typically to distributions that are affected by non geophysical phenomenon. However, we continue revisiting this criterion and probably in the next version of the product we will analyze the impact of not including this criterion of the skewness.*

Line 148. Where does the 25 come from?

We have clarified this in the text:

Lines 157-160:

*We discard specific salinity retrievals $s_n^{raw}(\gamma)$ when the corresponding SMOS debiased salinity anomaly ($s_n'(\gamma)$) is larger than $\sigma_{\gamma}$. Since we want to keep the geophysical variability, we include a threshold defined by $5\sigma_{\varphi,\lambda}$ being $\sigma^2_{\varphi,\lambda}$ the expected geophysical variance of the salinity at the gridpoint $(\varphi,\lambda)$. This is new with respect to the criterion proposed in (Olmedo et al., 2017). We discard the salinity retrievals that satisfy:*

Line 169. This is problematic. it is an assumption the authors are making, but it is not clear it is true. Can they please provide a reference or some other justification.

We have included the following plot in Figure 1:

[Figure]

The plot represents the mean difference between a constant salinity field, the annual climatology WOA13 and the salinity provided by Argo floats. As observed in the figure, the hypothesis is confirmed up to hundredths of psu.
We have included this in the text:
Lines 188-191:
*We use the constant annual reference WOA13 to assess this assumption. The top plot in Figure A1 shows the temporal evolution of the mean difference between the salinity field provided by WOA13 and the collocated uppermost salinity measurements provided by Argo floats. The results show that this hypothesis is true up to hundreds of psu.*

Figure A1. It's interesting that the difference decreases over time. Can the authors interpret this?
SMOS mission is actually an old mission. Some drifts have been detected at the level of brightness temperature measurements. There is no reference to this. This is actually a current topic in the internal meetings of the SMOS Payload Calibration Meetings.

Line 232. "spatial radio"?
We have changed:
"spatial radio" by "smoothing windows of radius"

Lines 291-292. Repeats from lines 205-210.

We have removed this sentence

Line 329. Missing ")"
Corrected

Line 358. "power-law" behaviour?
Corrected

Line 434. Repeats from line 415. Delete.
Deleted

Lines 456-459. "Figure A8..." This information is in the caption and does not need

to be repeated.
Deleted.

Line 497. Not remembering the earlier section where this is described... H_0 is the black curve in Fig. A12 and H-bar is the white one? Put this in the caption.
Yes. The reviewer is right. Thanks for notice! We have corrected the text and added the notation in the caption of the figure.

Line 690. This reference is undecipherable. Provide a URL. Ditto the Sabater and de Rosnay reference.
Corrected.

Section 5. It was not obvious how to access the data from the emodnet or cmems sites. Visualizations were available, but not the data themselves.
True. Both links correspond to visualization of the data. The access to the data is throughout the BEC SFTP service. Section 5 has been modified accordingly to make it clearer.

---

## Author Comment (AC2) · 30 Dec 2020

**Answers to Reviewer #2:**

This paper presents a new reprocessing of the BEC SMOS SSS global L3 and L4 products v2.0 for a 9-years period comprising 2011 to 2019, with an updated methodology with respect to previous BEC version. The updates concern new filtering criteria, a new empirical correction for latitudinal and seasonal biases, an improved scheme for preserving small-scale gradients close to the coast, and the provision of a sea surface salinity uncertainty. Three BEC products are presented : a high resolution product, a low resolution product (smoothed with a radius of 50km) and a L4 product produced using a multifractal fusion technique applied to SMOS L3 using OSTIA SST fields.

After the methodology presentation, evaluation exercises are performed :

-by comparing the new BEC products with Argo salinities over the whole time series

-by comparing the characteristics of the new BEC product relative to other satellite salinity products and to a CMEMS salinity product. These comparisons are restricted to the 2017 year. They concern not only comparisons to Argo but also comparisons of singularity exponents, of spectral properties and results obtained with triple collocations.

This is a huge piece of work by one of the two data centers delivering SMOS Level 3/4 products. Given the unique length of SMOS satellite salinity, it is a very important topic. The paper is clearly written and contains many interesting results. Nevertheless, I think several aspects would be worth to be deepened before publication, as this paper will serve as reference for future work:

Thank you very much for all your comments and suggestions. We think they have helped to improve the quality of the manuscript.

- The 2017 year is chosen to intercompare the various products because it is close to a normal year ('there is not any large-scale geophysical phenomenon'). But given that the BEC products are calibrated using a climatology and that the methodology makes use of many filterings, it would be important to evaluate the product skills relative to others during a non normal year, e.g. 2016.

We have included in the manuscript the comparison with Argo also for the year 2016. The other metrics are not expected to change very much with the selected year.

- The metrics used in the comparisons with Argo are mean, standard deviation and rms difference. They should also include R2 : indeed, the BEC methodology employs many filterings and R2 would allow to measure their efficiency in maintaining the SSS variability.

We have included R2 in the Tables containing the statistics with respect to Argo, they correspond to Tables A4 and A5 in the new version of the manuscript.

- By construction, the singularity exponents of the L4 product are expected to be

close to the ones of OSTIA SST so I don't think that comparing BEC L4 and OSTIA SST properties provides an independent validation of BEC L4 products. True. We have clarified this in the text. See the answer to the corresponding points below.

Detailed remarks : Abstract : not sure doi are useful in the abstract. This was suggested by the editor.

Abstract : last statement (iii) should be moderated (see below my comments about the triple collocation section) : I don't think the authors have an absolute reference allowing a measure of the error. Given the huge spatial variability in regions affected by rainfall and by continental freshwater discharges, I am surprised that a low resolution product is more accurate.

We have moderated this sentence by including that these are the results obtained from triple collocation:

Lines 19-21:

the results from triple collocation show the BEC SMOS level 4 product as the product with the lowest estimated salinity error in most of the global ocean and the BEC SMOS high-resolution level 3 as the one with the lowest estimated salinity error in regions strongly affected by rainfall and continental freshwater discharges.

In the section of triple collocation we have added a discussion, as suggested by the reviewer below (see also our answer in the corresponding point).

L20 : verb is missing Included.

L126 : radiometric sensitivities wording is confusing (see Randa et al (Recommended Terminology for Microwave Radiometry, National Institute of Standards and Technology Technical Note 1551 (August 2008) : Radiometric sensitivity is often used to mean radiometric resolution, but this use is discouraged in view of the definition of sensitivity).

I guess the authors mean radiometric resolution.

Thank you very much for providing us with this reference! We have changed the text accordingly.

Section 2.2.3 : It is useful to recall all these filterings. It might be useful to point out more specifically the changes with respect to Olmedo et al 2017 and to display maps of the number of filtered points.

We have included in the text which are the differences with respect to the filtering criteria previously applied: Lines 147-150: These filtering criteria are the same as the ones introduced in (Olmedo et al., 2017). The only difference is that now the criterion corresponding to the kurtosis is more relaxed: In (Olmedo et al., 2017) the set  $\{s_n^{raw}(gamma)\}$  was considered not valid and thus discarded out when the kurtosis of the distribution were larger than 4. Now we filter only platykurtotic distributions but not leptokurtotic ones.

Line 159:

This is new with respect to the criterion proposed in (Olmedo et al., 2017).

**Line 166-170:**

*Generally, in the open ocean,* \$\sigma\_{\varphi,\lambda}\$ is small, so equation and the previous filtering criteria have similar performances. However, in those with salinity dynamics, such reaions strona as coastal reaions, \$\sigma\_{\varphi,\lambda}\$ is not small and its contribution in equation 2 becomes dominant. Therefore, in those regions with strong salinity dynamics, the new filtering criteria is more relaxed and thus allows capturing better the salinity variability.

In particular, the distribution of the salinity is naturally skewed in rainy regions and in river plumes, what is the effect of such filterings in these regions ?

The main effect of this filtering criterion is to remove those acquisition conditions with historical time series regularly affected by RFI. In the following figure the values of the skewness for acquisition conditions at the center of the dwell line and at incidence angle values of 52 deg for ascending orbits are presented. The affectation of RFI close to the Euro-Asian coast is evident, besides the RFI tails close to Madagascar and the western coast of Africa. However, the reviewer is right in the point that the values of skewness lower than -1 close to the Northern Brazilian coast, are probably due to the advected plume of the Amazon by the North Brazilian current. We are currently revisiting this filtering criteria. In the regional products that we are developing in the framework of different ESA initiatives (Baltic+ Salinity dynamics and EO4SIBS projects) we are not applying any restriction on the skewness of the distributions. So, in future versions of the global product we will check the impact of not considering this filtering condition.

Skewness of the SSS distribution Acquistions at the center of the dwell line and incidence angle 52deg

-1,0 -0,6 -0,2 0,2 0,6 1,0

We have added a sentence to explain this in the manuscript: Lines 150-157:

Regarding the impact of the filtering criterion corresponding to the skewness, this is the same as the one proposed in Olmedo et al. (2017). This criterion aims at discarding ocean regions affected by RFI

contamination. Although some geophysical events tend to be not symmetric and fresh, as continental discharge and ice melting, and this leads to negative skewed salinity distributions, the typical skewness in these cases is around -0.5. The skewness values lower than -1 correspond typically to distributions that are affected by non geophysical phenomenon. However, we continue revisiting this criterion and probably in the next version of the product we will analyze the impact of not including this criterion of the skewness.

Equation 4 corresponds to a one sigma sorting, which seems very stringent, what is the rationale for this choice ? how sensitive is the result to this ?

At this point, we have salinity values that have been previously debiased and we collocate all of them in 9-days and a rectangular grid. We accumulate all the acquisitions, the acquisitions that we have per each incidence angle and per each dwell line, during the 9 days at a given lat-lon. For this set of debiased salinity values, we compute the mean and then, we apply a one-sigma criterion with respect to this mean. We have included the following sentence in the text: Lines 181-183:

This criterion was also applied in the previous version of the product. Since, at this step, the salinity retrievals are already debiased and they are temporally and spatially collocated, the criterion of one-sigma applied here is expected to reduce the noise of the level 3 salinity maps only.

Figure A2 : I suggest to add a figure displaying the polynomial correction. We have included the polynomial in Figure A2, as well as the description of the figure:

**Lines 216-218:**

In the bottom plots of Figure A2 the monthly interpolating polynomials p(m, varphi) are presented (in blue), as well as the mean difference  $Delta s_1(m, varphi, lambda)$  (in green). As observed in the plots, the approach of this correction has some limitations at high latitudes, where the sea ice dynamics also induce ice-sea contamination.

Section 2.2.6 : A major difficulty is how to filter SMOS retrieved salinity given that

salinity distribution is very skewed and that RFI contamination might lead to artificial skewed distribution too. In the updated methodology, more stringent filterings than in Olmedo et al. (2017) are applied.

The filtering criteria regarding the skewed distribution is the same as in Olmedo 2017. Only the filtering of kurtosis has been modified.

After having performed the serie of filterings and corrections, an inconsistency between the WOA reference and the mean corrected field appears in river plumes which is likely an effect of the skewed salinity distribution (Figure A3) but it seems that there is also a global north-south difference : could the authors refine the color scale of Figure A3 to allow a better display (e.g. with a 0.02psu resolution)?

In the open ocean differences with respect to the reference is about 0.04 psu.

We have investigated the causes of these discrepancies in the framework of the ESA regional initiatives Baltic+ Salinity Dynamics and EO4SIBS. In coastal regions, the main contribution to these discrepancies is because of the lat-lon resolution in which the climatological salinity distributions are computed. In the computation of the climatological representant that is used for the debiasing, we accumulate all the retrievals in a rectangular grid of 0.25°x0.25° in lat-lon and we use the eight neighbours of the given gridpoint to increase the statistics and thus obtain the central estimator of the distribution more accurate. However, the systematic errors close to the coast change rapidly in the eight neighbours and then this procedure is introducing inaccuracies in the estimation of the central estimator. In the open ocean, the order of the discrepancies is much smaller. We think that the reason for these discrepancies is probably due to the numerical truncations of the histograms we use in the climatological distributions (they are computed with salinity bins of 0.5 psu).